# *LiDAR*: Sensing Linear Probing Performance in Joint Embedding SSL Architectures

**Vimal Thilak, Chen Huang, Omid Saremi, Laurent Dinh, Hanlin Goh,
Preetum Nakkiran, Josh Susskind, Etai Littwin**
Apple
Correspondence to: {vthilak, elittwin}@apple.com

## Abstract

Joint embedding (JE) architectures have emerged as a promising avenue for acquiring transferable data representations. A key obstacle to using JE methods, however, is the inherent challenge of evaluating learned representations without access to a downstream task, and an annotated dataset. Without efficient and reliable evaluation, it is difficult to iterate on architectural and training choices for JE methods. In this paper, we introduce *LiDAR* (**Linear Discriminant Analysis Rank**), a metric designed to measure the quality of representations within JE architectures. Our metric addresses several shortcomings of recent approaches based on feature covariance rank by discriminating between informative and uninformative features. In essence, *LiDAR* quantifies the rank of the Linear Discriminant Analysis (LDA) matrix associated with the surrogate SSL task—a measure that intuitively captures the information content as it pertains to solving the SSL task. We empirically demonstrate that *LiDAR* significantly surpasses naive rank based approaches in its predictive power of optimal hyperparameters. Our proposed criterion presents a more robust and intuitive means of assessing the quality of representations within JE architectures, which we hope facilitates broader adoption of these powerful techniques in various domains.

> *"Measure what is measurable, and make measurable what is not so.."*
>
> - Galileo Galilei

## 1 Introduction

In recent years, self-supervised learning (SSL) has emerged as a pivotal technique for pretraining representations on extensive, unlabeled datasets, thus effectively alleviating the often burdensome labeling requirements (Chen et al., 2020; Assran et al., 2023; Chen & He, 2020; Caron et al., 2021; Bardes et al., 2021; Caron et al., 2018; 2020; Baevski et al., 2022; Zbontar et al., 2021; He et al., 2021; HaoChen et al., 2021; Grill et al., 2020). However, despite the remarkable progress made in SSL, assessing the quality of representations acquired through this method remains an open problem. This challenge is further amplified when considering *Joint Embedding* (JE) architectures, which notoriously suffer from uninterpretable loss curves that offer little clue to assess the progression of training. The conventional and widely adopted approach involves evaluating model performance by employing these representations in downstream tasks. Nevertheless, when the objective is to learn versatile representations applicable across diverse domains, this method demands substantial investments of time and resources to comprehensively evaluate performance across a multitude of tasks and datasets. Alternatively, limiting assessments to only a few datasets and tasks undermines confidence in the evaluation process.

As a result, a fundamental question arises: Can we effectively evaluate the quality of learned representations without relying on explicit downstream task evaluations? Addressing this inquiry necessitates the precise definition of "quality" in the context of representations. Subsequently, we must explore statistical estimators capable of quantifying this quality without depending on downstream evaluations. Beyond its theoretical implications, such a metric holds significant practical value, as it aids in model selection and the development of novel SSL algorithms.

Recent works (Garrido et al., 2022; Agrawal et al., 2022; Lu et al., 2023; Kalibhat et al., 2022) have introduced metrics that measure the quality of representations that correlate well with downstream tasks. Among these methods, the metrics introduced in (Garrido et al., 2022; Agrawal et al., 2022) are relevant to our work as they rely on statistics of the empirical feature covriance matrix. Notably, the recently introduced *RankMe* (Garrido et al., 2022) method shows that the rank of the feature covariance correlates surprisingly well with downstream performance, demonstrating SOTA results in label free hyperparameter selection. In this paper, we build upon *RankMe* by proposing a simple modification to the feature covariance matrix on which the rank is calculated, which significantly and consistently improves its predictive power of downstream performance. Our method, which we refer to as *LiDAR*, is motivated by a simple observation: the covariance spectrum can be easily and arbitrarily manipulated, and this manipulation is often encouraged by implicit or explicit regularizations in the SSL objectives. Consequently, a representation with a full rank covariance matrix may result from spurious factors rather than representing rich semantic features. As a result, even though methods such as *RankMe* demonstrate impressive results, we show that non-trivial additional gains can be had rather effortlessly. We summarize our contributions below:

1. We introduce *LiDAR*, a method for assessing representation quality of JE SSL objectives, and theoretically motivate it. *LiDAR* uses the SSL objective in its definition, providing a more intuitive and robust metric for assessing SSL representations.

2. We conduct a comprehensive set of experiments spanning multiple JE architectures, both transformer and resnet based. These include contrastive and regularized methods, as well as newer models such as I-JEPA and data2vec, which leverage masking techniques. We demonstrate that the *LiDAR* metric correlates significantly and consistently higher with downstream linear probing performance than *RankMe* as measured by both the Spearman Rank and Kendall rank correlation coefficient or Kendall's $\tau$.

3. We show that *LiDAR* demonstrates consistently strong performance in hyperparameter selection, outperforming *RankMe*. We further demonstrate this capability in randomized trials, where multiple hyperparameters are varied simultaneously.

## 2 PRELIMINARIES

### 2.1 SELF SUPERVISED LEARNING

The main goal of self supervised pre-training is to learn a general purpose data representation $f(x)$ that is transferable to a large variety of downstream tasks. Informally, the degree of transferability of $f$ can be measured by how easy it is to learn a downstream task given $f$. In practice, a set of downstream tasks $\{T_j\}$ are used to assess the transferability of $f$ by training additional readout networks $\phi_j$ on top of $f$. That is, a successful pre-training scheme involves finding $f$ such that there exists "simple" functions $\phi_j$, such that $\phi_j \circ f$ solves $T_j$ to a reasonable degree, and $\phi_j$ are "easy" to learn [1]. In the popular linear probing protocol, a linear readout functions $\phi_j$ is used to assess the quality of $f$. For the remainder of the paper we restrict our investigations to linear probing.

### 2.2 JOINT EMBEDDING ARCHITECTURES

A common paradigm in the current practice of SSL is the multi-view setting. In its simple form, a pair of inputs are processed by a JE architecture, where each input is encoded separately by a (possibly shared) encoder. The SSL objective is then tasked with learning a representation such that "compatible" pairs that share semantic information are easily predictive of the each other, perhaps with the help of a latent variable. Various methods in the literature fall under this general category, which mostly differ by how compatible inputs are sampled, and how they avoid representational collapse. We can formally define a JE architecture with an encoder [2] $f(x) : \mathcal{X} \rightarrow \mathbb{R}^d$ and a projector function [3] $\psi(f) : \mathbb{R}^d \rightarrow \mathbb{R}^p$. We adopt the terminology in (Garrido et al., 2022), and refer to the output of the encoder $f$ as a *representation*, and the output of the composed encoder

---

[1] We use the terms "simple" and "easy to learn" here loosely as requiring few samples

[2] The encoder function need not be shared between views, however for simplicity and without loss of generality we assume it is

[3] Some SSL methods do not use a projector function, in which we can assume it is the identity function

and projector $e = \psi \circ f$ as an *embedding*. A common theme among most JE SSL objectives is that they seek to learn embedding functions that produce similar embeddings for compatible views, which are typically generated by various forms of data augmentation. However, more recent JE architecture somewhat depart from this paradigm by using input masking, and introducing latent variables in the projector function $\psi$. For example, in I-JEPA (Assran et al., 2023) and data2vec (Baevski et al., 2022), $\tilde{x}$ and $x$ represent partially masked and unmasked inputs respectively, and $\psi(f; z)$ is tasked with predicting the parts of the representation $f(x)$ given spatial locations provided by $z$. For simplicity and without loss of generality, we remove the explicit notation $z$ in the definition of $\psi$. Note that, absent any input reconstruction loss, JE architectures potentially promote more abstract representations by filtering fine grained pixel level information. However, without an input reconstruction term, the loss used is often uninformative of the actual metric of interest, which is the zero or few shot transfer of the learned representation to downstream tasks. Compounding this challenge, extended training durations can significantly deteriorate representation quality, even when employing a set of hyperparameters with proven performance, as illustrated in Figure 1. In such cases, finding an appropriate early stopping criterion is critical.

**Dimensional Collapse**  JE architectures are prone to various forms of representation dimension collapse, which can manifest in different flavors (Ziyin et al., 2022; Jing et al., 2021; Hua et al., 2021). In contrastive methods, dimensional collapse occurs when learning results in an excessively low dimensional representations in a way that hinders downstream performance. Regularized methods, when insufficiently regularized either implicitly or explicitly, can theoretically suffer complete collapse, where the learned representation trivializes to a constant. However, beyond such a rough categorization of the training process to partial or complete collapse, recent literature alludes to a more nuanced observation that, in some settings, the feature covariance eigenspectrum can be used to accurately gauge the quality of the learned representation as it pertains to downstream performance.

### 2.2.1 Embeddings and Feature Covariance Eigenspectrum

Recent developments in the evaluation of JE-SSL representations have introduced methods that leverage information derived from the spectrum of either the representations or the embeddings (RankMe, Garrido et al. (2022) or from the spectrum of the covariance matrix, $\alpha$-Req, Agrawal et al. (2022)). Garrido et al. (2022) who propose a new measure named *RankMe* and demonstrate that this measure, when applied across different hyperparameter configurations, exhibits a strong correlation with downstream performance across various settings and tasks. In order to define *RankMe*, we start with a dataset $\{x_i\}_{i=1}^n$ drawn iid from some input distribution $\mathcal{D}$, and an embedding function $e(x) \in \mathbb{R}^d$ that produces an embedding matrix $\mathbf{Z}$ with dimensions $(n, d)$ whose singular values are denoted as $\sigma = \sigma_1, \sigma_2, \ldots \sigma_{\min(n,d)}$. These singular values reveal insights into the nature of the mapping function $f(x)$. RankMe (Garrido et al., 2022) introduced a soft measure of effective rank expressed as $\exp(-\sum_{i=1}^p p_i \log p_i)$, where $p_i = \frac{\sigma_i}{\|\sigma\|_1} + \epsilon$, and $\epsilon$ represents a small constant.

In a related work, $\alpha$-Req (Agrawal et al., 2022) uses insights from infinite dimensional spaces to argue that the eigenspectrum of representations, i.e., eigenspectrum of the feature covariance should decay at an ideal rate of $\lambda_i \sim O(i^{-1})$ where $\lambda_i$ is the $i^{th}$ eigenvalue of the covariance matrix. However, it's essential to note that any condition imposed on the eigenspectrum alone may not be sufficient to ascertain the representational power of the mapping function $f$. This limitation arises from the fact that a simple linear mapping, such as $f(x) = Wx$ with a weight matrix $W$, can manipulate the covariance matrix's spectrum arbitrarily. Additionally, even a random mapping could exhibit a high effective rank without necessarily translating into significant downstream performance improvements. Notably, (Li et al., 2022) showed that the loss value and covariance spectrum can be used in tandem to predict performance. However, their method requires training a classifier on offline data, making it highly inefficient as an unsupervised method. *LiDAR* follows a similar intuition with a more efficient and effective formulation: A representation with a high effective rank, when coupled with a low objective loss, points to successful training. To balance the two terms we leverage a discriminative method from classical statistics, repurposed to SSL settings.

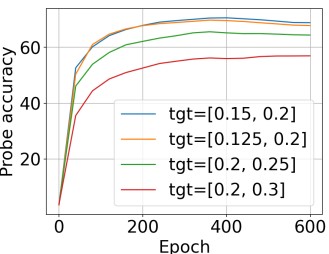 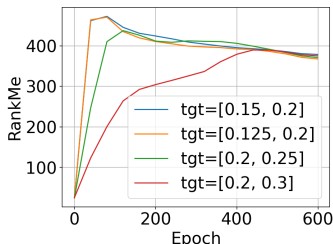 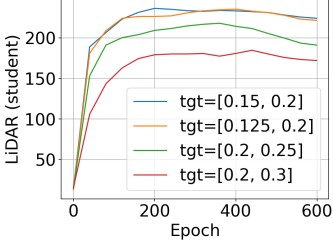

Figure 1: ViT-Base architecture trained with I-JEPA (Assran et al., 2023) by varying the target mask scale hyperparametr. We observe that (1) *RankMe* correlates poorly with downstream performance for most models, with the exception of the worst performing model, and (2) *LiDAR* correlates highly with downstream performance for all models.

## 3 METHOD

Our method is based on *linear discriminant analysis* (Bishop, 2007), a classical label-aware dimensionality reduction and classification algorithm which we adopt to the multi-view setting common in the SSL literature. Absent downstream task and labels, we use clean samples as surrogate classes. For an input distribution $\mathcal{D}$, consider a generic JE architecture with an embedding function $e$. For any clean input $x$, let $\mathcal{D}_x$ denote a conditional distribution over all transformed [4] inputs given $x$. Define $\mu_x = \mathbb{E}_{\tilde{x} \sim \mathcal{D}_x}[e(\tilde{x})]$ and $\mu = \mathbb{E}_{x \sim \mathcal{D}}[\mu_x]$. we define the generalized covariance $\Sigma_{\text{lidar}}$ by:

$$\Sigma_{\text{lidar}}(e) = \Sigma_{\text{w}}(e)^{-\frac{1}{2}} \Sigma_b(e) \Sigma_{\text{w}}(e)^{-\frac{1}{2}} \tag{1}$$

where $\Sigma_b(e) = \mathbb{E}_{x \sim \mathcal{D}} \left[ (\mu_x - \mu)(\mu_x - \mu)^\top \right], \Sigma_{\text{w}}(e) = \mathbb{E}_{x \sim \mathcal{D}} \mathbb{E}_{\tilde{x} \sim \mathcal{D}_x} \left[ (e(\tilde{x}) - \mu_x)(e(\tilde{x}) - \mu_x)^\top \right] + \delta I_p$ where $\delta$ is a small positive constant, and $I_p$ is the identity matrix of dimension $p$. Let $\lambda = \lambda_1, ..., \lambda_p$ be the eigenvalues of $\Sigma_{\text{lidar}}$, then *LiDAR* is defined by applying the smooth rank measure introduced in (Roy & Vetterli, 2007) on $\Sigma_{\text{lidar}}$ :

$$\textit{LiDAR}(e) = \exp\Big( -\sum_i p_i \log p_i \Big), \quad p_i = \frac{\lambda_i}{\|\lambda\|_1} + \epsilon \tag{2}$$

where $\epsilon$ is a small positive constant. In practice, we use unbiased estimates of $\Sigma_w, \Sigma_b$ using hyperparameters $n, q$ where $n$ is the numbers of surrogate classes (clean samples), and $q$ is the number of transformed samples per class. Note that, as in classical LDA, the eigenvalues $\lambda_1, ..., \lambda_p$ measure variance along discriminative directions. Hence, *LiDAR(e)* takes into account the SSL objective that is being optimized, and ignores directions of variability in $e$ that are useless in solving the SSL task. We postulate the existence of such directions, as JE techniques implicitly or explicitly incorporate measures to preserve representations from collapsing. In simpler terms, some variations in $e$ may not necessarily arise from high-level semantic features driven by the SSL objective, but rather from arbitrary attributes influenced by regularization. This phenomenon can clearly be seen in I-JEPA, where the rank of a representation inflates early on in training, only to rapidly diminish, with peak downstream performance occurring far from the point of maximal rank (see Figure 1). This makes any standard data covariance rank based measures extremely limited in their ability to spot the point of optimal performance in a training run. In contrast, the *LiDAR* method initiates at a much lower rank and steadily ascends, more closely aligning with downstream linear probing performance, as depicted in Figure 1. We delve deeper into the theoretical rationale behind *LiDAR* in Appendix Section 8, where we prove a proposition which illustrates the effect of uninformative information on the *LiDAR* metric.

## 4 EXPERIMENTAL DETAILS

*LiDAR* is intended to serve as a proxy metric to compare the quality of representation as they relate to downstream performance. This task is challenging due to the inherent uncertainty of the nature

---

[4]Data augmentations, or otherwise data points which are treated as positive samples

of the downstream task. In this paper, we focus on downstream classification tasks, employing the widely adopted linear probing protocol. A robust metric in this sense is one that consistently replicates the downstream performance ranking observed with a ground truth oracle across a set of pre-trained models. As in existing solutions, we only compare models from the same class, each one pre-trained using a different set of predefined hyperparameters. It is important to emphasize that a linear correlation between the metric and the oracle is not implied, hence we resort to established statistical tests which assess the strength and direction of the monotonic relationship between two variables. In other words, we seek to measure how well the relationship between two variables, the metric and the oracle, can be described using a monotonic function. We note that such evaluations go beyond simply picking the best model according to the metric of interest out of a set of models, due to possible outliers in the set.

**Spearman's rank correlation coefficient**   The Spearman correlation coefficient (Spearman, 1987; 1961) is essentially the Pearson correlation coefficient computed on the ranked values of two variables. While Pearson's correlation assesses linear relationships, Spearman's correlation evaluates monotonic relationships, which can be linear or non-linear in nature. In cases where there are no duplicate data values, a perfect Spearman correlation of either +1 or -1 indicates that each of the variables exhibits a flawless monotonic relationship with the other, forming a perfect monotone function. Given two sequences of real numbers $X = x_1, ..., x_n$ and $Y = y_1, ..., y_n$ and their corresponding ranking $R(X), R(Y)$, the Spearman correlation coefficient is given by:

$$r_s = 1 - \frac{d \sum_i (R(x_i) - R(y_i))^2}{n(n^2 - 1)} \qquad (3)$$

**Kendall's Tau rank correlation coefficient**   The Kendall's Tau correlation coefficient (Kendall, 1938b;a) is another popular alternative the to Spearman rank correlation, and is known to be superior when the sample size is small and has many tied ranks. The Kendall's Tau coefficient uses concordant and discordant pairs in its measure. For indices $j > i$, the pairs $\{x_i, x_j\} \in X, \{y_i, y_j\} \in Y$ are said to be concordant if the sort order of both pairs agree, otherwise they are said to be discordant. Let $C$ and $D$ denote the number of concordant and discordant pairs in $X, Y$ then the Kendall's $\tau$ is given by $|C - D|/(C + D)$.

**Top ranking model**   In similar spirit to (Garrido et al., 2022), as an additional evaluation we report the top ranking model given a metric of interest, and its downstream performance, and compare it to the optimal model according to the oracle. This protocol can be seen as a noisy estimate of Kendall's Tau rank correlation coefficient.

## 4.1   MODELS, HYPERPARAMETERS AND DATA

We use 5 different multiview JE SSL methods, spanning contrastive and regularized methods, as well as ResNet and transformer based. We use I-JEPA (Assran et al., 2023) and data2vec (Baevski et al., 2022) as representative of more recent masking based approaches that are not reliant on domain specific data augmentation. We use SimCLR (Chen et al., 2020) as a representative of contrastive methods, while we use DINO (Caron et al., 2021) as an example of self-distillation method and VICReg (Bardes et al., 2021) as representatives of regularized methods. Note that I-JEPA and data2vec use transformer based encoders by design. We use a vision transformer (ViT) (Dosovitskiy et al., 2021) based encoder for DINO as well, and ResNet-50 (He et al., 2016) based encoders for SimCLR and VICReg. We vary different hyperparameters per method. The varied hyperparameters range from optimization related ones such as learning rate, and weight decay, architecture specific hyperparameters such as softmax temperature, and data augmentation and masking based hyperparameters. We stress that some SSL objectives such as I-JEPA and data2vec, which rely on specific forms of input masking, are extremely sensitive to the the masking hyperparameters, hence providing an important testbed for LiDAR. We highlight the drawback of conducting a grid search over a single hyperparameter, due to the fact that all the remaining frozen ones are typically highly optimized to the task, offering a somewhat contrived testbed. Hence, in addition to a standard grid search, we use random search over all hyperparameters. This is done by uniformly sampling each hyperparameter from a fixed range, providing a better cover for the space. We use the Imagenet-1k dataset (Russakovsky et al., 2015) for all experiments. We use the train split as the source dataset

| Hyperparameter | RankMe | RankMe (aug.) (Student) | LiDAR (Student) | RankMe (aug.) (Teacher) | LiDAR (Teacher) |
|---|---|---|---|---|---|
| Learning rate | 0.6835 | 0.7829 | **0.8443** | 0.6734 | 0.7506 |
| Weight decay | 0.5040 | 0.7034 | **0.7331** | 0.4792 | 0.5278 |
| Target mask scale | 0.2867 | 0.6786 | **0.7937** | 0.2956 | 0.2927 |
| Context mask scale | 0.4246 | 0.7867 | **0.8482** | 0.3929 | 0.5556 |
| Overall | 0.5830 | 0.7513 | 0.8159 | 0.5713 | 0.6828 |

Table 1: I-JEPA: Kendall's $\tau$ coefficient between effective ranks of RankMe, RankMe (aug.) and LiDAR and linear probe accuracy. Hyperparameters are varied via grid search.

| | RankMe | RankMe (aug.) (Student) | LiDAR (Student) | RankMe (aug.) (Teacher) | LiDAR (Teacher) |
|---|---|---|---|---|---|
| Random search | 0.8314 | **0.8989** | 0.8434 | 0.8487 | 0.8616 |

Table 2: I-JEPA: Kendall's $\tau$ coefficient between effective ranks of *RankMe*, *RankMe (aug.)* and *LiDAR* and linear probe accuracy. Hyperparameters are varied via random sampling.

for pretraining and linear probing, and use the test split as the target dataset. For each pretrained checkpoint, we train a linear probe on the train split, which we denote as the oracle, and record its test performance on the test split.

## 4.2 IMPLEMENTATION CONSIDERATIONS

The implementation of LiDAR entails computing empirical approximations to $\Sigma_w, \Sigma_b$, which differs from one SSL method to another due to the inherent differences in the way input pairs are sampled. As a general rule, for each SSL method we use its own input transformations without alterations. In asymmetrical architectures, we compare both branches, denoted as the "student" and the "teacher" for evaluation. In transformer based architectures (as employed by design in I-JEPA, data2vec) we pool the final embedding/representation to produce vectors. For methods such as I-JEPA and data2vec, computing the *RankMe* score on the embeddings is not a straightforward task. This is due to the fact that the projector function's task in both does not serve as a representation expander, rather it serves as a conditional predictor, predicting masked representations, conditioned on the spacial locations of the masked patches. For these methods, we evaluate *RankMe* on the student or teacher encoder instead. For SimCLR and VICReg, we copy the implementation details from (Garrido et al., 2022). The hyperparameters used to train and evaluate the models are listed in Appendix 9. Finally, in our analysis within the context of *RankMe*, we have noticed that utilizing the same data augmentations for both training and feature covariance matrix computation consistently leads to improved performance. In our experiments, we provide empirical results for both "vanilla" and augmented *RankMe* as baselines.

| Metric | LR | WD | Target mask scale | Context mask scale | Overall |
|---|---|---|---|---|---|
| ImageNet Oracle | 70.5800 | 70.5800 | 70.5800 | 70.5800 | 70.5800 |
| RankMe | 56.7580 | 60.2040 | 60.2040 | 59.8960 | 56.7580 |
| RankMe (aug.) (Student) | **67.8680** | **67.8680** | **67.8680** | **67.8680** | **67.8680** |
| RankMe (aug.) (Teacher) | 56.7580 | 52.6720 | 50.4200 | 59.8960 | 56.7580 |
| LiDAR (Student) | **67.8680** | **67.8680** | **67.8680** | **67.8680** | **67.8680** |
| LiDAR (Teacher) | 65.1080 | 64.1920 | 65.5820 | 67.5420 | 65.1080 |

Table 3: I-JEPA: Linear probe accuracy recovered by *RankMe* and *LiDAR* on ImageNet-1K dataset. Hyperparameters set via grid search.

| ImageNet Oracle | RankMe | RankMe (aug.) (Student) | RankMe (aug.) (Teacher) | LiDAR (Student) | LiDAR (Teacher) |
|---|---|---|---|---|---|
| 68.9840 | 53.3700 | 53.3700 | 53.3700 | 61.1760 | **66.8880** |

Table 4: I-JEPA: Linear probe accuracy recovered by *RankMe* and *LiDAR* on ImageNet-1K dataset. Hyperparameters set via random sampling.

### 4.3 COMPUTE CONSIDERATIONS

While calculating the *LiDAR* score is a straightforward and computationally efficient procedure, it does introduce an additional computational cost when compared to standard covariance estimation. This additional cost arises from the need to perform matrix inversion in $\Sigma_w^{-0.5}$. It can be, in principle, time-consuming when dealing with high-dimensional embeddings. In our experiments, we have observed that this computational overhead is generally inconsequential for all tested models. The cost tends to be dominated by the computational demand of the feature extraction's forward process. Nonetheless, we note the following trivial bound on the rank of $\Sigma_{\text{lidar}}$:

$$\text{Rank}(\Sigma_{\text{lidar}}) \leq \min\big(\text{Rank}(\Sigma_w), \text{Rank}(\Sigma_b)\big) \tag{4}$$

Since $\text{Rank}(\Sigma_b)$ is bounded by the number of surrogate classes used $n$ (number of "clean" samples used to generate augmented samples), simple dimensionality reduction can be used when $p >> n$ to reduce the rank of $\Sigma_w$ before inversion, without any loss in performance. We find in our experiments that $n = 1k, q = 50$ is sufficient to saturate performance of I-JEPA, data2vec and DINO methods. Our evaluations with SimCLR requires $n = 5k, q = 10$ while we use $n = 10k, q = 10$ for VICReg as these methods provide much longer features compared to I-JEPA, data2vec and DINO.

## 5 RESULTS

As strikingly evident from our experimental results, the *LiDAR* metric correlates surprisingly well with downstream performance, as measured by linear probing. In the vast majority of experiments, we see a significant improvement over *RankMe* in terms of the Kendall's $\tau$ and Spearman's rank correlation to the oracle, and an improved performance in hyperparameter selection. Due to space constraints, we defer the reader to the Appendix 9 for a comprehensive view of experimental details, supplementary empirical results, and additional figures. In the main text, we have included a representative selection to provide an overview.

Tables 1 and 3 present a comprehensive analysis of the results obtained for the I-JEPA. The evaluation involves a comparison between *LiDAR*, assessed on both the teacher and student branches, and *RankMe*, with and without data augmentation, alongside the oracle reference. We observe a general trend where *LiDAR* applied on the student branch correlates much higher with the oracle than the teacher branch. Overall we observe significant outperformance of *LiDAR* over *RankMe* applied on clean and augmented inputs for all hyperparameters tested. A noteworthy observation from Table 3 is that *RankMe*, when coupled with data augmentation to compute the feature covariance matrix, can match *LiDAR*'s performance for the best-selected model, albeit trailing significantly when data augmentation is not employed. Table 2 and Table 4 present results from randomized trials where hyperparameters are randomly sampled within predefined ranges. In these trials, a consistent trend of *LiDAR* outperforming both versions of *RankMe* is evident, as can also be seen in Figure 2. Tables 5 and 6 extend our analysis to the data2vec model, and the findings parallel those observed for I-JEPA. Notably, *LiDAR* consistently selects more performant hyperparameters than both variants of *RankMe*, narrowly missing oracle-level performance.

Table 7 summarizes our results for VICReg (Bardes et al., 2021). This table additionally reports metrics calculated at different checkpoints throughout the training process. Across all checkpoints (epochs 20, 40, 60, 80, 100), we consistently observe significantly higher correlations with the oracle performance. It's worth highlighting that, with VICReg, *LiDAR* achieves optimal results when applied to the representation rather than the embedding, as embedding-based evaluations result in dramatic performance degradation, a phenomenon aligning with the non-monotonic relationship between rank and performance reported by (Garrido et al., 2022). This illustrates that high rank is a necessary but not a sufficient condition for high performance.

| Hyperparameter | RankMe | RankMe (aug.) (Student) | LiDAR (Student) | RankMe (aug.) (Teacher) | LiDAR (Teacher) |
|---|---|---|---|---|---|
| Learning rate | 0.2410 | 0.3419 | **0.5077** | 0.3172 | 0.4716 |
| Mask ratio | 0.2683 | 0.2381 | **0.4657** | 0.2195 | 0.4170 |
| Overall | 0.2238 | 0.2799 | **0.5531** | 0.2626 | 0.5227 |

Table 5: data2vec: Kendall's $\tau$ for data2vec between effective ranks of RankMe, RankMe (aug.) and LiDAR and linear probe accuracy.

| Metric | LR | Mask ratio | Overall |
|---|---|---|---|
| ImageNet Oracle | 60.3920 | 55.9780 | 60.3920 |
| RankMe | 48.6040 | 48.6980 | 48.6980 |
| RankMe (aug.) (Student) | 48.6040 | 51.4500 | 51.4500 |
| LiDAR (Student) | **59.3720** | **52.7460** | **59.3720** |
| RankMe (aug.) (Teacher) | 48.6040 | 51.4500 | 51.4500 |
| LiDAR (Teacher) | **59.3720** | **52.7460** | **59.3720** |

Table 6: data2vec: Linear probe accuracy recovered by *RankMe* and *LiDAR* with data2vec on ImageNet-1K dataset.

Further validation of *LiDAR*'s efficacy in hyperparameter selection is provided in Appendix Table 10, where the method achieves results that are within a fraction of a percentage point from oracle-level performance across all considered hyperparameters. Table 8 shifts the focus to SimCLR, a widely used contrastive self-supervised learning method. The evaluation centers on the final checkpoint obtained after 100 training epochs. Table 8b reveals that *LiDAR* consistently demonstrates the highest correlation among the three metrics under scrutiny. Moreover, Table 8a confirms that *LiDAR* consistently selects the most optimal hyperparameters among the three methods considered. Lastly we also study the behavior of *LiDAR*. Table 16 lists the results for a ViT-Small trained with DINO on ImageNet-1K dataset. We observe that *LiDAR* evaluated with both the teacher and student branches show stronger correlation than *RankMe*.

## 6 LIMITATIONS

While we observe *LiDAR* significantly improves upon *RankMe* in most experiments, it is important to emphasize its drawbacks. Notably, we have observed instances where the *LiDAR* metric exhibits a negative correlation with probe accuracy, particularly pronounced in scenarios like VICReg when dealing with higher dimensional embeddings. This phenomenon underscores the intrinsic complexity of the relationship between rank, however it is measured, and downstream task performance.

| Epoch | RankMe | RankMe (aug. dataset) | LiDAR |
|---|---|---|---|
| 20 | 0.6081 | 0.6466 | **0.6957** |
| 40 | 0.4315 | 0.5949 | **0.8699** |
| 60 | 0.4065 | 0.5381 | **0.8809** |
| 80 | 0.3904 | 0.5905 | **0.9032** |
| 100 | 0.3174 | 0.5209 | **0.9161** |

| Epoch | RankMe | RankMe (aug. dataset) | LiDAR |
|---|---|---|---|
| 20 | 0.4476 | 0.4718 | **0.5323** |
| 40 | 0.2984 | 0.4597 | **0.7097** |
| 60 | 0.2702 | 0.3750 | **0.7218** |
| 80 | 0.2823 | 0.4435 | **0.7823** |
| 100 | 0.2056 | 0.3790 | **0.8105** |

(a) VICReg: Spearman Rank coefficient  (b) VICReg: Kendall's $\tau$ coefficient

Table 7: VICReg: Correlation between effective rank estimated by *RankMe* and *LiDAR* and probe accuracy evolution during training. Each row corresponds to a checkpoint collected at epoch specified in the table.

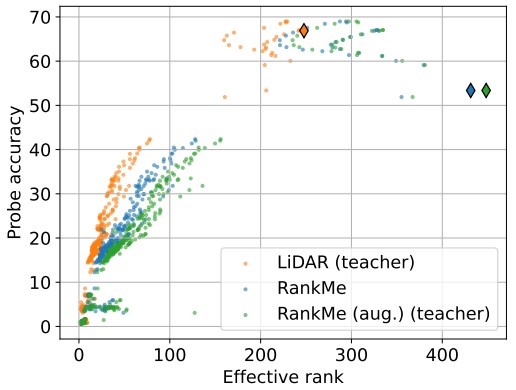 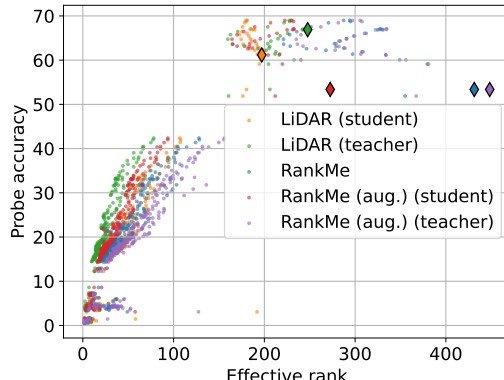

(a) Linear probe accuracy for *RankMe*, teacher-based *LiDAR* and teacher-based augmented-*RankMe*.

(b) Linear probe accuracy for all *LiDAR*, augmented-*RankMe* variants and vanilla *RankMe*.

Figure 2: I-JEPA: ViT-Base architecture trained on Imagenet-1K with 20 sets of hyperparameters drawn uniformly at random. Each point represents a checkpoint while the ⋄ marker represents the model selected by each metric. We observe that (1) *LiDAR* outperforms *RankMe* (see Table 2), (2) augmented-*RankMe*, our variant of *RankMe*, shows the highest correlation (see Table 2), and (3) remarkably *LiDAR* shows the best hyperparameter selection performance by recovering the highest downstream performance.

| Metric | LR | WD | Temp. | Overall |
|---|---|---|---|---|
| Imagenet Oracle | 58.2370 | 56.6740 | 57.4710 | 59.1420 |
| *RankMe* | 55.8950 | 55.1490 | 56.0390 | 56.4630 |
| *RankMe* (aug. dataset) | 57.2010 | 55.8170 | 56.3170 | 57.8260 |
| *LiDAR* | **57.8940** | **56.3020** | **57.0920** | **58.9270** |

(a) Linear probe accuracy.

| Correlation | RankMe | RankMe (aug. dataset) | LiDAR |
|---|---|---|---|
| Kendall's $\tau$ | 0.4982 | 0.5761 | **0.8167** |

(b) Kendall's $\tau$ correlation coefficient.

Table 8: SimCLR: (a) Linear probe accuracy recovered by *RankMe*, augmented-*RankMe* and *LiDAR* on ImageNet-1K dataset at the end of training and (b)Kendall's $\tau$ correlation coefficient between effective rank estimated by *RankMe* and *LiDAR* and probe accuracy per hyperparameter.

These two factors are not necessarily causally linked; a high rank does not guarantee superior performance. It is essential to acknowledge the computational overhead associated with calculating the *LiDAR* metric, which often requires high-dimensional matrix inversion. This increases the method's overall computational cost as shown in Appendix 12 using I-JEPA as an example. Consequently, the feasibility of incorporating *LiDAR* as a loss signal for pretraining should be carefully considered, as it could be prohibitively expensive to naively evaluate it at each iteration during the training process. Due to the sheer volume of experiments and necessary compute, we focus solely on linear probing in this work. We expect *LiDAR*'s impressive performance to carry over to nonlinear probing protocols as well. Lastly, it's worth noting that our approach is contingent on the sampling strategy for positive pairs $x$ and $\tilde{x}$ which varies among methods, making cross-method comparisons a challenge.

# 7    CONCLUSION

We introduced *LiDAR*, a novel approach to evaluate self-supervised learning models. It builds upon the foundation laid by the *RankMe* metric. Through a series of experiments, we have demonstrated *LiDAR*'s superiority over *RankMe*, a representative covariance rank-based approach. It enables accurate and label-independent self-supervised learning assessment of learned representations. Our method is a powerful tool for practitioners seeking to efficiently optimize their models in data-scarce environments. As for future work, investigating *LiDAR* as a possible loss term is a particularly compelling one. This would require a different formulation to make the method more SGD-compatible. We anticipate *LiDAR*'s integration into standard SSL toolkits that has the potential to reshape model evaluation and have large impact on the future advancements of the field.

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

## 8 THEORETICAL MOTIVATION

In this section we illustrate a heuristic as to why we might expect *LiDAR* to outperform previous methods relying on the covariance spectrum. As an exemplary JE SSL method, we consider the VICreg objective, which is comprised of three terms:

$$\mathcal{L}_{\text{VIRreg}} = \underbrace{\lambda\mathcal{L}_{\text{inv}}}_{\text{Invariance}} + \underbrace{\mu\mathcal{L}_{\text{var}} + \nu\mathcal{L}_{\text{cov}}}_{\text{Regularization}} \tag{5}$$

where $\lambda, \mu, \nu$ are hyperparameters, and $\mathcal{L}_{\text{inv}}, \mathcal{L}_{\text{var}}, \mathcal{L}_{\text{cov}}$ are the invariance, variance and covariance terms computed over a minibatch. In this objective, the regularization term which is comprised of the variance and covariance terms explicitly encourages the embedding function's covariance to be of high rank, while the invariance term insures that compatible pairs are mapped to similar embeddings. We highlight that a low regularization loss is achievable by random embeddings, which might be high rank, but devoid of any utility as for downstream tasks. A measure of representation quality which is based on covariance rank alone would therefore, theoretically, fail to capture this failure case, as it only pertains to the regularization term. Indeed, balancing the hyperparameters $\lambda, \mu, \nu$ is necessary to prevent superficially inflating the covariance rank. On the other hand, *LiDAR* is invariant to information that is not used to discriminate between surrogate classes in the SSL objective. To make this point concrete, consider a (centered) embedding function $e(x) : \mathbb{R}^d \to \mathbb{R}^p$, a random independent noise vector $\mu \in \mathbb{R}^r$ such that $\mathbb{E}[\mu] = \mathbf{0}, \mathbb{E}[\mu\mu^\top] = \Sigma_\mu \in \mathbb{R}^{r \times r}$, and consider the (random) embedding functions $\tilde{e}(x) = [e(x)^\top, \mu^\top]^\top : \mathbb{R}^d \to \mathbb{R}^{p+r}$. Naturally, when measuring the downstream performance of $e, \tilde{e}$, we expect that $e$ should not be worse than $\tilde{e}$. (in practice performance is measure on the representation $f$, however for the sake of a clean argument we will ignore this technicality). In turn, this should, ideally, translate to $LiDAR(\tilde{e}) \leq LiDAR(e)$. *LiDAR* (unlike covariance spectrum based approaches) indeed captures this property, as illustrated in the following proposition:

**Proposition 1.** *Let $\mathcal{D}$ denote a distribution over inputs $x \in \mathbb{R}^d$, let $\mathcal{D}_x$ denote a conditional distribution of transformed inputs given $x$. Let $\lambda = \lambda_1, ..., \lambda_p$ be the eigenvalues of $\Sigma_{lidar}(e)$. Assume that $\frac{\|\lambda\|_\infty}{\|\lambda\|_1} < 1 - \exp[-1]$ and set constants $\epsilon, \delta$ such that $\epsilon < 1 - \frac{\|\lambda\|_\infty}{\|\lambda\|_1}$, and $\delta < (\exp[-1] - \epsilon)\|\lambda\|_1$. Then, it holds that:*

$$\text{LiDAR}(\tilde{e}) \leq \text{LiDAR}(e) \exp\left[-2p\log\left(\frac{\|\lambda\|_1}{\|\lambda\|_1 + r\delta}\right) - \frac{r\delta}{\|\lambda\|_1}\log\left(\frac{\delta}{\|\lambda\|_1}\right)\right] \tag{6}$$

*Proof.* From the independence of the noise vector $\mu$, it is easy to see that:

$$\Sigma_b(\tilde{e}) = \begin{pmatrix} \Sigma_b(e) & \mathbf{0} \\ \mathbf{0} & \mathbf{0} \end{pmatrix}, \quad \Sigma_w(\tilde{e}) = \begin{pmatrix} \Sigma_w(e) & \mathbf{0} \\ \mathbf{0} & \Sigma_\mu \end{pmatrix}, \quad \Sigma_{\text{lidar}}(\tilde{e}) = \begin{pmatrix} \Sigma_{\text{lidar}}(e) & \mathbf{0} \\ \mathbf{0} & \delta I_r \end{pmatrix} \tag{7}$$

Then, we have that:

$$LiDAR(\tilde{e}) = \exp\left[-\sum_{i=1}^p \left(\frac{\lambda_i}{\|\lambda\|_1 + r\delta} + \epsilon\right)\log\left(\frac{\lambda_i}{\|\lambda\|_1 + r\delta} + \epsilon\right) + \mathcal{R}\right] \tag{8}$$

where:

$$\mathcal{R} = \exp\left[-r\left(\frac{\delta}{\|\lambda\|_1 + r\delta} + \epsilon\right)\log\left(\frac{\delta}{\|\lambda\|_1 + r\delta} + \epsilon\right)\right] \tag{9}$$

$$\leq \exp\left[-\frac{r\delta}{\|\lambda\|_1}\log\left(\frac{\delta}{\|\lambda\|_1}\right)\right] \tag{10}$$

where we used the fact that $\frac{\delta}{\|\lambda\|_1 + r\delta} + \epsilon < \exp[-1]$ by assumption to deduce the maximum of the function $|x\log(x)|$ in the interval $x \in (0, \exp[-1])$. Note that:

$$\exp\left[-\sum_{i=1}^p \left(\frac{\lambda_i}{\|\lambda\|_1 + r\delta} + \epsilon\right)\log\left(\frac{\lambda_i}{\|\lambda\|_1 + r\delta} + \epsilon\right)\right] \tag{11}$$

$$= \exp\left[-\sum_{i=1}^p \left(\frac{\lambda_i}{\|\lambda\|_1 + \delta r} + \epsilon\right)\log\left(\frac{\lambda_i}{\|\lambda\|_1} + \epsilon\right)\right. \tag{12}$$

$$\left. -\sum_{i=1}^p \left(\frac{\lambda_i}{\|\lambda\|_1 + \delta r} + \epsilon\right)\log\left(\frac{\frac{\lambda_i}{\|\lambda\|_1 + r\delta} + \epsilon}{\frac{\lambda_i}{\|\lambda\|_1} + \epsilon}\right)\right] \tag{13}$$

Since $\forall_i,\ \frac{\lambda_i}{\|\lambda\|_1} + \epsilon < 1$ by assumption, we have:

$$\exp\Big[ - \sum_{i=1}^{p} \Big( \frac{\lambda_i}{\|\lambda\|_1 + \delta r} + \epsilon \Big) \log \Big( \frac{\lambda_i}{\|\lambda\|_1} + \epsilon \Big) \Big] \tag{14}$$

$$\leq \exp\Big[ - \sum_{i=1}^{p} \Big( \frac{\lambda_i}{\|\lambda\|_1} + \epsilon \Big) \log \Big( \frac{\lambda_i}{\|\lambda\|_1} + \epsilon \Big) \Big] = \textit{LiDAR}(e) \tag{15}$$

hence we can write:

$$\exp\Big[ - \sum_{i=1}^{p} \Big( \frac{\lambda_i}{\|\lambda\|_1 + r\delta} + \epsilon \Big) \log \Big( \frac{\lambda_i}{\|\lambda\|_1 + r\delta} + \epsilon \Big) \Big] \tag{16}$$

$$\leq \textit{LiDAR}(e) \exp\Big[ - p\Big(1 + \epsilon\Big) \min_i \log \Big( \frac{\frac{\lambda_i}{\|\lambda\|_1 + r\delta} + \epsilon}{\frac{\lambda_i}{\|\lambda\|_1} + \epsilon} \Big) \Big] \tag{17}$$

$$\leq \textit{LiDAR}(e) \exp\Big[ - 2p \log \Big( \frac{\|\lambda\|_1}{\|\lambda\|_1 + r\delta} \Big) \Big] \tag{18}$$

Combining equations 10 and 18, we get the result:

$$\textit{LiDAR}(\tilde{e}) \leq \textit{LiDAR}(e) \exp\Big[ - 2p \log \Big( \frac{\|\lambda\|_1}{\|\lambda\|_1 + r\delta} \Big) - \frac{r\delta}{\|\lambda\|_1} \log(\frac{\delta}{\|\lambda\|_1}) \Big] \tag{19}$$

$$\square$$

Note that an immediate consequence of Proposition 1 is that for $\delta << \|\lambda\|_1$ we have that $\textit{LiDAR}(\tilde{e}) \leq \textit{LiDAR}$.

**Eigenspectrum Decay**   As additional motivation, we invoke the arguments made in (Stringer et al., 2019) and later expanded in (Agrawal et al., 2022) on the optimal eigenspectrum decay rate in an infinite dimensional embedding space, given by $\lambda_i \sim \Theta(i^{-1})$. In (Stringer et al., 2019), it was shown that a slower decay rate would necessarily imply a non-smooth kernel function, which, in turn, implies a non-monotonic relationship between rank and downstream performance. *LiDAR* circumvents this issue by implementing whitening operation through the inverse of $\Sigma_w$. This, in theory, permits the attainment of both a smooth embedding and the flexibility for eigenvalue decay in $\Sigma_{\text{LDA}}$ to occur at a very gradual rate. It is essential to acknowledge, however, that while the smoothness can be maintained, an excessively high LDA rank may, in theory, have adverse consequences on downstream performance, owing to other underlying factors. We, therefore, defer a more comprehensive theoretical exploration of the implications of *LiDAR* to future research endeavors.

## 9 IMPLEMENTATION DETAILS

### 9.1 VICREG

VICReg (Bardes et al., 2021) proposes an explicit regularization to prevent dimensional collapse in self-supervised learning methods. VICReg (Bardes et al., 2021) consists of the standard invariance term, a variance term to encourage networks to not suffer from dimensional collapse and a covriance term that encourages the covariance matrix of embeddings to be approximately diagonal. Let $z_i \in \mathbb{R}^d$ denote the features from a neural network. The variance term is given by:

$$var(Z) = \frac{1}{d} \sum_{j=0}^{d-1} \max\left(0, 1 - S\left(z_j, \epsilon\right)\right) \tag{20}$$

where $S\left(x, \epsilon\right)$ denotes the standard deviation and $\epsilon$ is a small value to prevent numerical instability. The covariance term is given by:

$$c(Z) = \frac{1}{d} \sum_{i \neq j} [C(Z)]_{i,j}^2 \tag{21}$$

while the invariance term is the $L_2$ distance between the usual positive data pairs used to optimize the network. The complete loss function used to optimize a network in VICReg is given by:

$$L\left(Z, Z'\right) = \frac{\lambda}{n} \sum_i \|z_i - z_j\| + \mu\left[var\left(Z\right) + var\left(Z'\right)\right] + \nu\left[c\left(Z\right) + c\left(Z'\right)\right] \tag{22}$$

where $\lambda$, $\mu$ and $\nu$ are hyperparameters that control the contribution from the invariance, standard deviation and covariance terms respectively.

We use the reference implementation [5] provided by Bardes et al. (2021) to train a ResNet-50 (He et al., 2016) backbone on Imagenet-1K dataset (Russakovsky et al., 2015). The projector used is a standard multi-layer perceptron (MLP) with dimensions 8192-8192-8192. We train a ResNet-50 with VICReg for 100 epochs using hyperparameters and training protocol described in (Bardes et al., 2021). The trained backbone is probed by trianing a classifier on the frozen features of the backbone. We use a setup that is described in *RankMe* including data preprocessing and optimizer-related hyperparameters to train a probe for 30 epochs. We use 32 hyperparametr sets that include described in Garrido et al. (2022) in our experiments.

We use ImageNet-1K (Russakovsky et al., 2015) training data as our source dataset for self-supervised learning (SSL). We use 10000 images from the source dataset, i.e., ImageNet-1K training split and 10 augmentations per image to construct a labeled dataset needed to calculate *LiDAR* and augmented-*Rankme*. We use 25600 images from the source dataset (ImageNet-1K) to calculate *RankMe* as done by Garrido et al. (2022). All pipelines that calculate various metrics employ the exact data augmentation pipeline used for SSL pretraining. The results for VICReg are as follows:

- We calculate the *LiDAR*, *RankMe*, augmented-*RankMe* for the final checkpoint after SSL training. Figure 3 shows a scatter plot of the 32 checkpoints where we plot the various effective rank metrics versus probe accuracy.

- We calculate the effective rank metrics on checkpoints collected every 20 epochs during training. Figure 4 shows the evolution of the metrics during training.

- Table 9 shows the correlations estimated via Spearman rank correlation and Kendall's $\tau$ correlation tests for all checkpoints collected during training.

---

[5]https://github.com/facebookresearch/vicreg

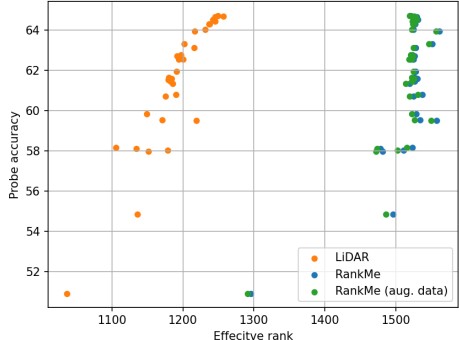

Figure 3: VICReg: Performance of VICReg representations measured by RankMe and LiDAR. Each point refers to a checkpoint evaluated after 100 epochs of self-supervised pretraining. LiDAR shows strong correlation

| Correlation | RankMe | RankMe (aug. dataset) | LiDAR |
|---|---|---|---|
| Spearman rank | 0.3174 | 0.5209 | **0.9161** |
| Kendall's $\tau$ | 0.2056 | 0.3790 | **0.8105** |

Table 9: VICReg: Compare RankMe and LiDAR using Spearman Rank correlation and Kendall's $\tau$ correlation measures after 100 epochs of training. VICReg representations are used to estimate *RankMe* and *LiDAR* for the 32 hyperparameter sets considered in our experiments. The hyperparamter values are identical to the values considered by (Garrido et al., 2022)

| Metric | cov. | inv. | LR | WD |
|---|---|---|---|---|
| Imagenet Oracle | 64.7380 | 62.7800 | 63.9500 | 61.6500 |
| *RankMe* | 64.5400 | 59.5400 | **63.9500** | **59.5200** |
| *RankMe* (aug. dataset) | 64.5400 | 59.5400 | **63.9500** | **59.5200** |
| *LiDAR* | **64.7080** | **62.5720** | **63.9500** | **59.5200** |

Table 10: VICReg: Linear probe accuracy recovered by *RankMe*, augmented-*RankMe* and *LiDAR* on ImageNet-1K dataset at the end of training. The metrics presented above are calculated with representations

| Epoch | RankMe | RankMe (aug. dataset) | LiDAR |
|---|---|---|---|
| 20 | 0.4476 | 0.4718 | **0.5323** |
| 40 | 0.2984 | 0.4597 | **0.7097** |
| 60 | 0.2702 | 0.3750 | **0.7218** |
| 80 | 0.2823 | 0.4435 | **0.7823** |
| 100 | 0.2056 | 0.3790 | **0.8105** |

(a) Kendall's $\tau$ correlation coefficient

| Metric | cov. | inv. | LR | WD |
|---|---|---|---|---|
| Imagenet Oracle | 64.7380 | 62.7800 | 63.9500 | 61.6500 |
| *RankMe* | 64.5400 | 59.5400 | **63.9500** | **59.5200** |
| *RankMe* (aug. dataset) | 64.5400 | 59.5400 | **63.9500** | **59.5200** |
| *LiDAR* | **64.7080** | **62.5720** | **63.9500** | **59.5200** |

(b) Linear probe accuracy recovered by *RankMe*.

Table 11: VICReg: (a) Kendall's $\tau$ correlation coefficient between effective rank estimated by *RankMe* and *LiDAR* and probe accuracy per hyperparameter and (b) Linear probe accuracy recovered by *RankMe*, augmented-*RankMe* and *LiDAR* on ImageNet-1K dataset at the end of training. The metrics presented above are calculated with representations.

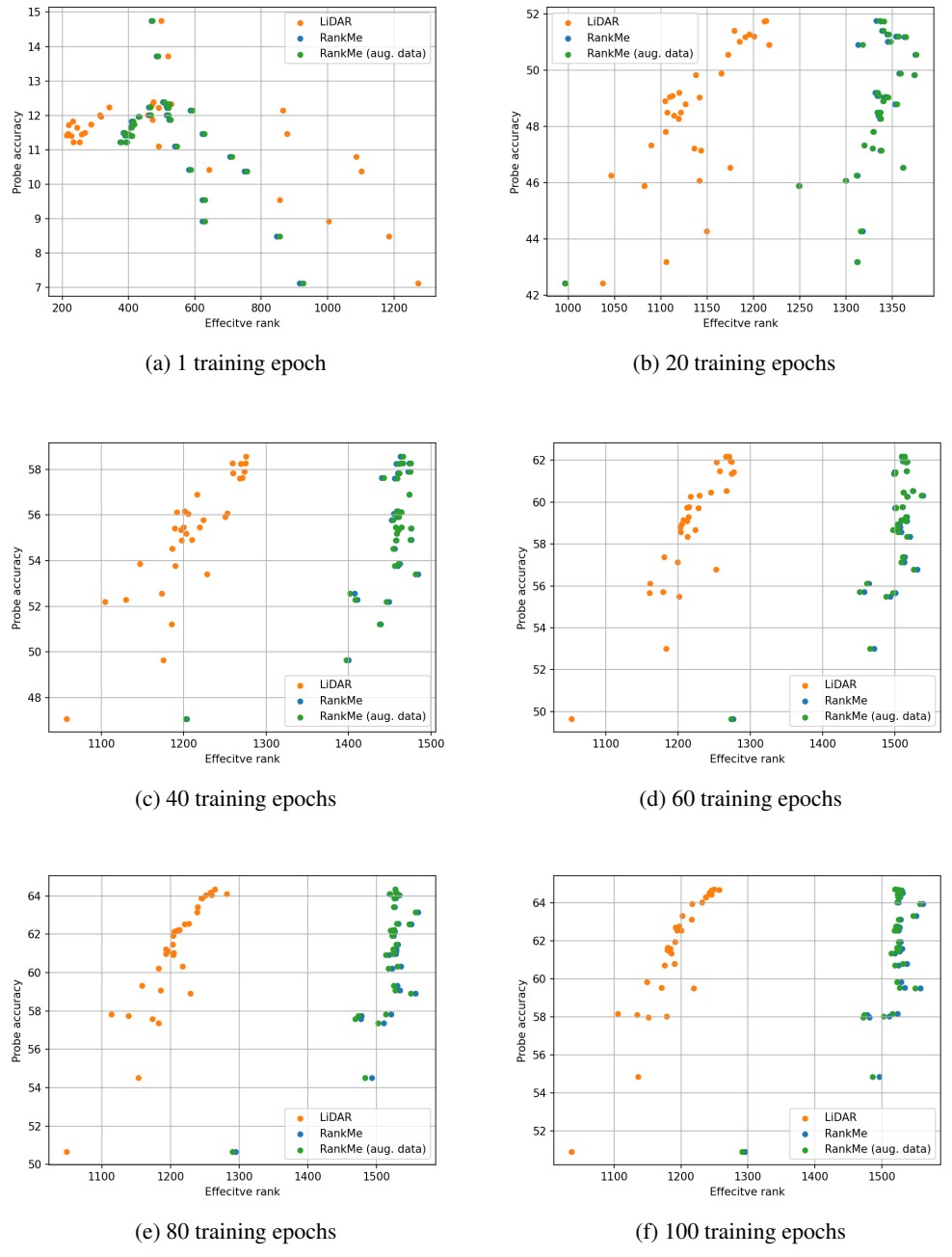

Figure 4: VICReg: Performance of VICReg representations measured by RankMe and LiDAR. Each point represents a row of hyperparameters among the 32 sets of hyperparameters consdiered in our experiments. The hyperparamter values are identical to the values considered by (Garrido et al., 2022)

## 9.2 I-JEPA

Image-based Joint-Embedding Predictive Architecture (I-JEPA) (Assran et al., 2023) is a recently proposed non-generative approach to learn semantically strong representations in a self-supervised manner. I-JEPA splits an image into a context block and several target blocks and uses the context block to predict the target blocks. Note that each block is composed of image patches. The core innovation in I-JEPA is the design of a masking strategy that is shown to lead to semantically strong representations when used in conjunction with Vision Transformers (ViTs) (Dosovitskiy et al., 2021). I-JEPA uses a ViT to encode the non-masked context patches and another ViT to predict the encodings for the masked out target patches. The target representations are provided by a target encoder which is an exponential moving average (EMA) version of the context encoder. In this work we refer to the context encoder as the student encoder and the target encoder as the teacher and use these terms interchangeably in our presentation. The loss function is applied to the embeddings that are output by the student and the teacher and is given by:

$$\frac{1}{M} \sum_{n=1}^{M} \sum_{i \in B_i} \|y_i - \hat{y}_i\|^2 \tag{23}$$

where $M$ denotes the number of target blocks, $B_i$ denotes a set of block indices in a target and $y$ and $\hat{y}$ denote the embeddings provided by the student and the teacher respectively. The asymmetry introduced due to student and teacher encoders allows I-JEPA to avoid representation collapse (Assran et al., 2023).

In order to apply *LiDAR* for I-JEPA Assran et al. (2023) we create a labeled dataset by first selecting a fixed number of images at random and treat each image as a class. We then apply the masking approach proposed in I-JEPA (Assran et al., 2023) to create multiple instances of a class to create a labeled dataset needed to calcualte the linear discriminant analysis matrix for *LiDAR*. The embeddings $y$ and $\hat{y}$ described in 23 are used as inputs to estimate Student and Teacher *LiDAR* measures. We use the output of the student encoder to calculate *RankMe* measure. As described in Section 4.2 the embeddings $y$ and $|haty$ are used to calculate the augmented *RankMe* metric which is denoted as *RankMe* (aug.) in the results. We use 1000 images and 50 augmentations to construct the labeled dataset for *LiDAR* and augmented-*RankMe* while we use 10000 image samples for *RankMe*. Self-supervised training is run for 600 epochs with an effective batch size of $2048$ using the training protocol described in I-JEPA (Assran et al., 2023). The downstream task consists of linear probing frozen representations on the ImageNet-1K dataset (Russakovsky et al., 2015). The probe is optimized with Adam (Kingma & Ba, 2015) optimizer for 20 epochs with a starting learning rate of 0.01 and a step learning rate schedule where the base learning rate is dropped by a factor 10 after 15 epochs. The following hyperparamter sets are used in our experiments to empirically estimate the performance of *RankMe* augmented-*RankMe* and *LiDAR*:

- Learning rate from $0.001, 0.002, 0.004, 0.006$ and $0.008$. Figure 5, Figure 6 show the results of experiments where we vary the learning rate parameter alone. Table 12 and Table 1 show the rank correlation coefficients while Table 3 show the accuracy recovered by *RankMe* and *LiDAR*.

- Weight decay from $0.05, 0.1, 0.2$ and $0.4$. Figure 7, Figure 8 show the results of experiments where we vary the weight decay parameter alone. Note that the weight decay is kept fixed throughought training in this set of experiments. Table 12 and Table 1 show the rank correlation coefficients while Table 3 show the accuracy recovered by *RankMe* and *LiDAR*.

- Target mask scale factor from $[0.15, 0.2], [0.125, 0.2], [0.2, 0.25]$ and $[0.2, 0.3]$. Figure 9, Figure 10 show the results of experiments where we target mask scale ratio alone. Table 12 and Table 1 show the rank correlation coefficients while Table 3 show the accuracy recovered by *RankMe* and *LiDAR*.

- Target mask scale factor from $[0.85, 1.0], [0.75, 1.0], [0.65, 1.0]$ and $[0.4, 1.0]$. Figure 11, Figure 12 show the results of experiments where we target mask scale ratio alone. Table 12 and Table 1 show the rank correlation coefficients while Table 3 show the accuracy recovered by *RankMe* and *LiDAR*.

Additionally, we conduct a random hyperparameter search experiment where we sample 20 sets of hyperparamters uniformly and train a ViT-B (Dosovitskiy et al., 2021) with I-JEPA (Assran et al., 2023). The hyperparametrs were sampled from the following uniform distributions:

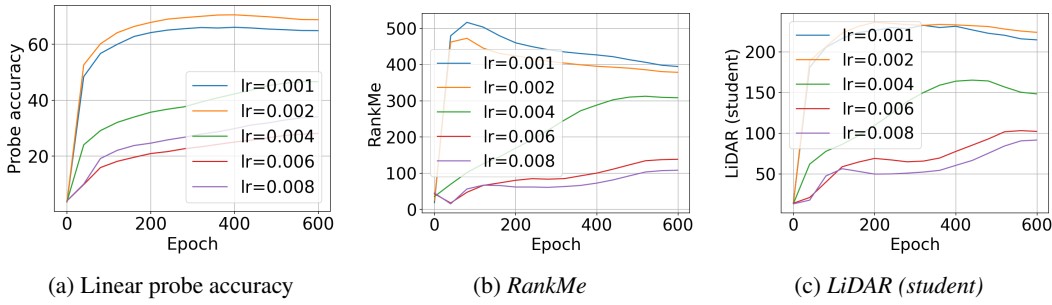

(a) Linear probe accuracy        (b) *RankMe*        (c) *LiDAR (student)*

Figure 5: I-JEPA: ViT-Base architecture trained on Imagenet-1K by varying the learning rate. Plots show the evolution of metrics over training time.

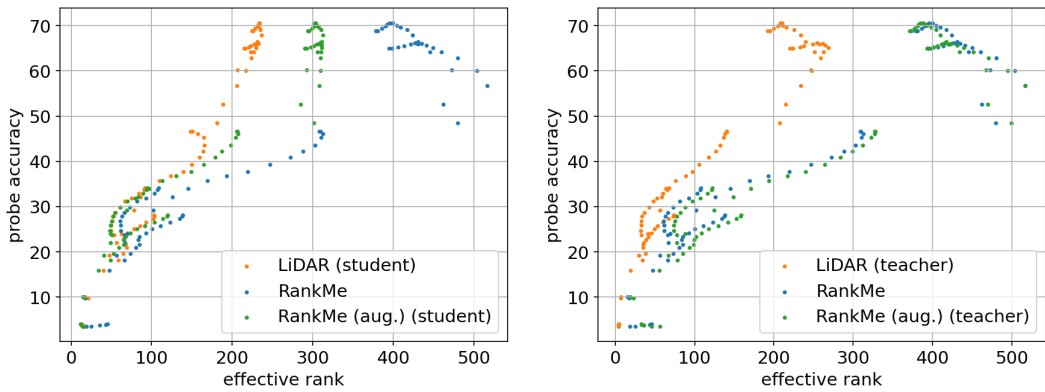

Figure 6: I-JEPA: Scatter plot of checkpoints collected during training of ViT-Base architecture on Imagenet-1K by varying the learning rate.

- learning rate from $[0.000125 \, 0.0002]$
- weight decay from $[0.05 \, 0.4]$
- target scale (minimum) from $[0.1 \, 0.2]$
- target scale (maximum) from $[0.2 \, 0.4]$
- context scale (minimum) from $[0.3 \, 0.95]$

The results of these experiments are shown in Figure 2 and the correlations are available in Table 2 and Table 13. The probe accuracy recovered for hyperparameters generated via random search are shown in Table 4.

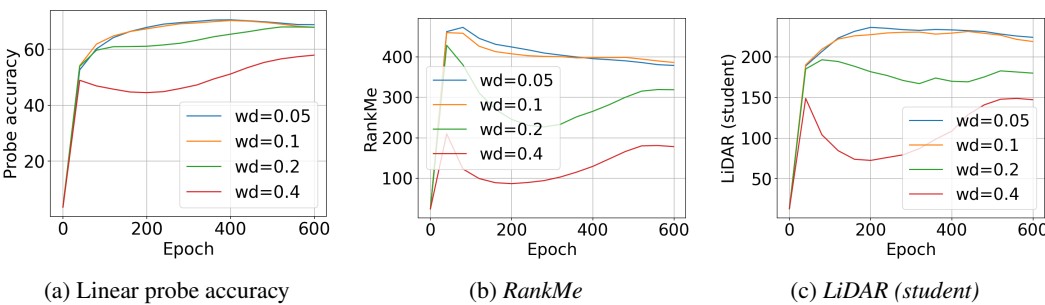

(a) Linear probe accuracy        (b) *RankMe*        (c) *LiDAR (student)*

Figure 7: I-JEPA: ViT-Base architecture trained on Imagenet-1K by varying the weight decay. Plots show the evolution of metrics over training time.

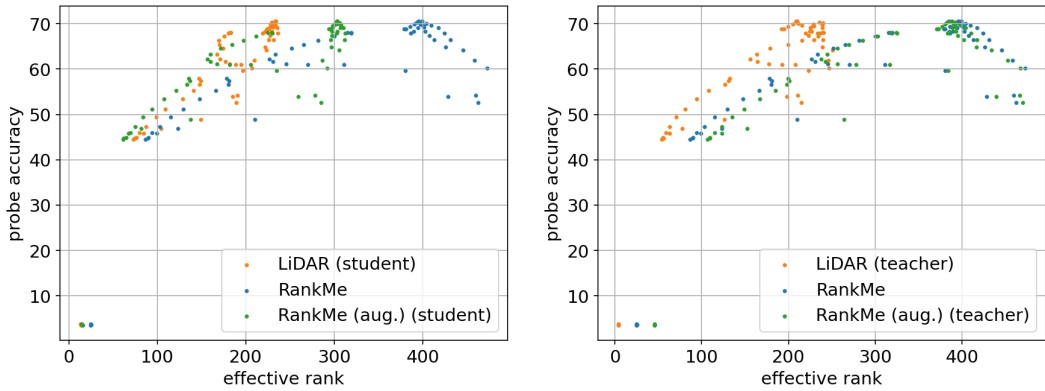

Figure 8: I-JEPA: Scatter plot of checkpoints collected during training of ViT-Base architecture on Imagenet-1K by varying weight decay.

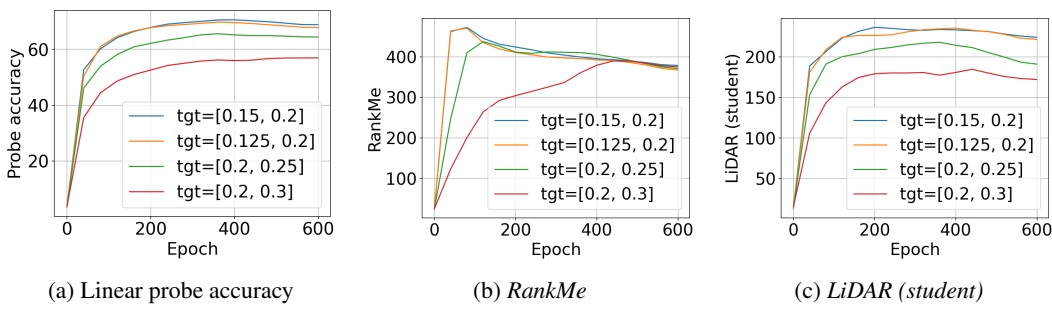

(a) Linear probe accuracy

(b) *RankMe*

(c) *LiDAR (student)*

Figure 9: I-JEPA: ViT-Base architecture trained on Imagenet-1K by varying the target mask scale hyperparameter. Plots show the evolution of metrics over training time.

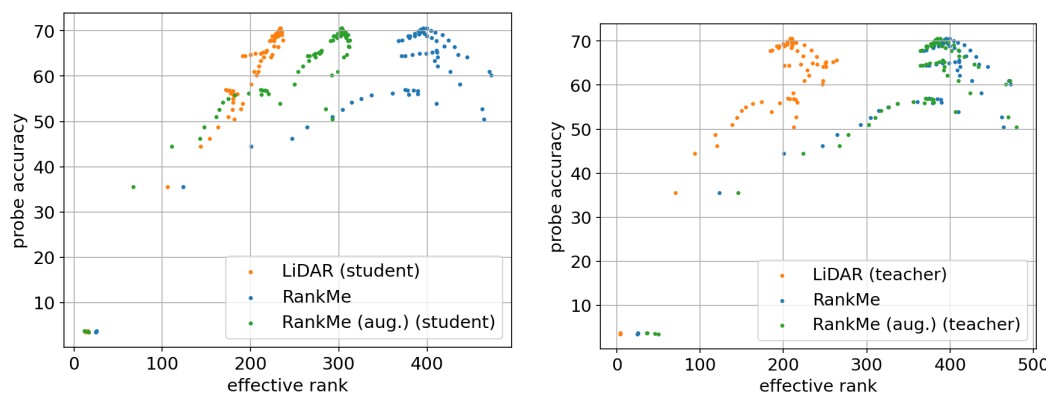

Figure 10: I-JEPA: Scatter plot of checkpoints collected during training of ViT-Base architecture on Imagenet-1K by varying the target mask scale factor.

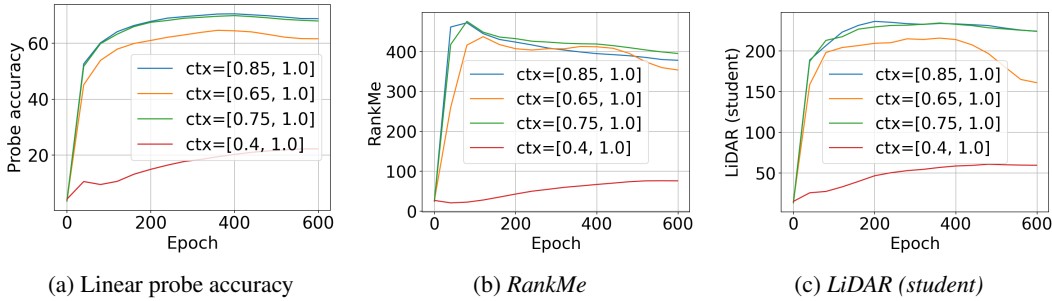

(a) Linear probe accuracy          (b) *RankMe*          (c) *LiDAR (student)*

Figure 11: I-JEPA: ViT-Base architecture trained on Imagenet-1K by varying the context mask scale hyperparameter. Plots show the evolution of metrics over training time.

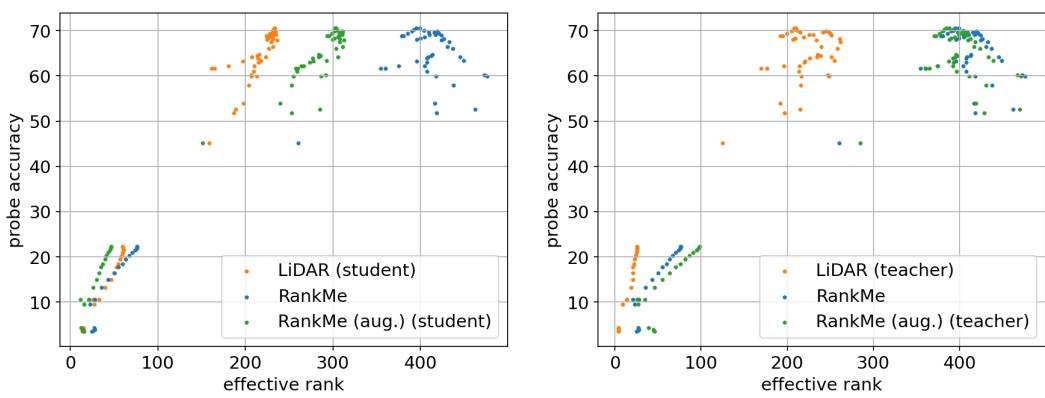

Figure 12: I-JEPA: Scatter plot of checkpoints collected during training of ViT-Base architecture on Imagenet-1K by varying the context mask scale factor.

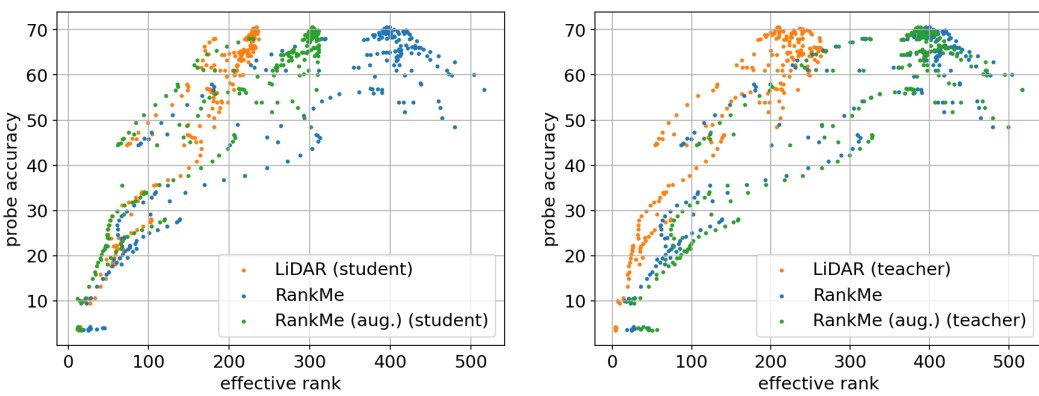

Figure 13: I-JEPA: Aggregated scatter plot of checkpoints collected during training of ViT-Base architecture on Imagenet-1K. Hyperparameters varied are one of learning rate, weight decay, target mask scale or context mask scale. Plots show that effective rank estimated by *LiDAR* has high correlation with linear probe accuracy.

| Hyperparameter | RankMe | RankMe (aug.) (Student) | LiDAR (Student) | RankMe (aug.) (Teacher) | LiDAR (Teacher) |
|---|---|---|---|---|---|
| Learning rate | 0.8775 | 0.9258 | **0.9605** | 0.8747 | 0.9030 |
| Weight decay | 0.6381 | 0.8669 | **0.8884** | 0.6196 | 0.6978 |
| Target mask scale | 0.4008 | 0.8429 | **0.9407** | 0.4069 | 0.3851 |
| Context mask scale | 0.5800 | 0.9180 | **0.9590** | 0.5533 | 0.6940 |
| Overall | 0.7700 | 0.9104 | **0.9494** | 0.7627 | 0.8475 |

Table 12: I-JEPA: Spearman rank correlation coefficient for I-JEPA between effective ranks of RankMe, RankMe (aug.) and LiDAR and linear probe accuracy.

| | RankMe | RankMe (aug.) (Student) | LiDAR (Student) | RankMe (aug.) (Teacher) | LiDAR (Teacher) |
|---|---|---|---|---|---|
| Random search | 0.9470 | **0.9770** | 0.9511 | 0.9486 | 0.9722 |

Table 13: I-JEPA: Spearman rank correlation coefficient between effective ranks of *RankMe*, augmented-*RankMe* and *LiDAR* and linear probe accuracy. Hyperparmaeters are generated via random sampling.

## 9.3 DATA2VEC

data2vec (Baevski et al., 2022) is a self-supervised learning approach that aims to predict latent representations of input data based on a masked view of the input data. The idea behind data2vec (Baevski et al., 2022) is similar to I-JEPA (Assran et al., 2023) but the details of masking and the predictor used to predict embeddings are different.

We follow the same protocol described for I-JEPA above in terms of constructing a labeled dataset for *LiDAR* and augmented-*RankMe* with 10000 samples from ImageNet-1K and apply 50 augmentations. The augmentations applied in this case are identical to the augmentation described in (Baevski et al., 2022) and available via a reference implementation provided by the authors [6]. We use 10000 source images to calculate *RankMe*. We train a ViT-B (Dosovitskiy et al., 2021) model for 800 epochs with an effective batch size of 2048 and with other hyperparameters described in (Baevski et al., 2022). During training, we save a checkpoint every 40 epochs for analysis and downstream linear probing. Linear probing is done on frozen representations of the encoder using LARS optimizer (You et al., 2017) with a base learning rate of 0.01 scaled via a linear scaling rule with a base batch size of 256 (Goyal et al., 2018) and an effective batch size of 16384 samples. We consider the following SSL training hyperparameters in our experiments to test the performance of *LiDAR* and *RankMe* and its augmented variant:

- Learning rate from 0.0007, 0.001, 0.002 and 0.004. Figure 14 and Table 14 and Table 5 show the results from these experiments
- Set number of mask patches to either 120 (default) or 80 while we fix the learning rate to 0.0007. Figure 16, Table 14 and Table 5 show the results from these experiments
- Table 15 shows the probe accuracy recovered by the various effective rank metrics.

---

[6]https://github.com/facebookresearch/data2vec_vision/tree/main/beit

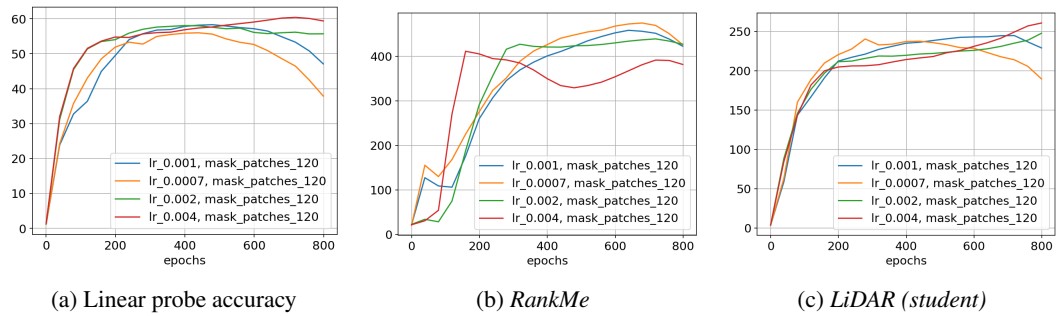

(a) Linear probe accuracy      (b) *RankMe*      (c) *LiDAR (student)*

Figure 14: data2vec: ViT-Base architecture trained on Imagenet-1K by varying the learning rate. Plots show the evolution of metrics over training time.

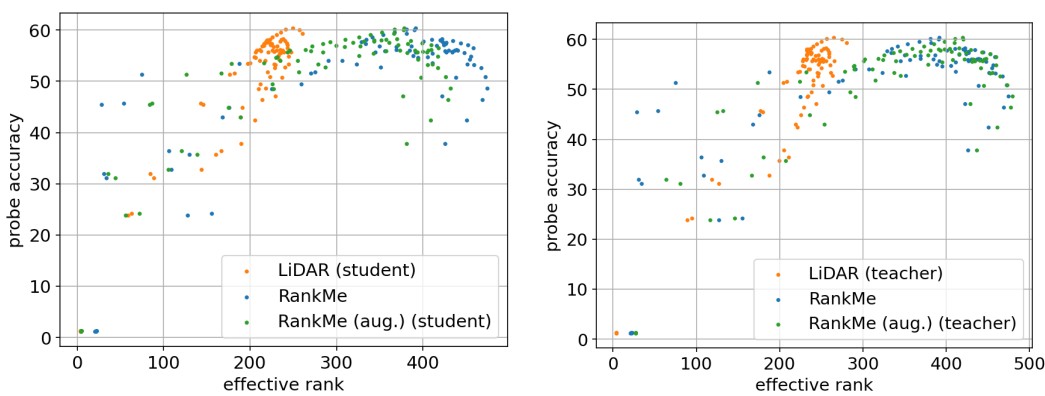

Figure 15: data2vec: Scatter plot of checkpoints collected during training of ViT-Base architecture on Imagenet-1K. The learning rate was varied in this experiment.

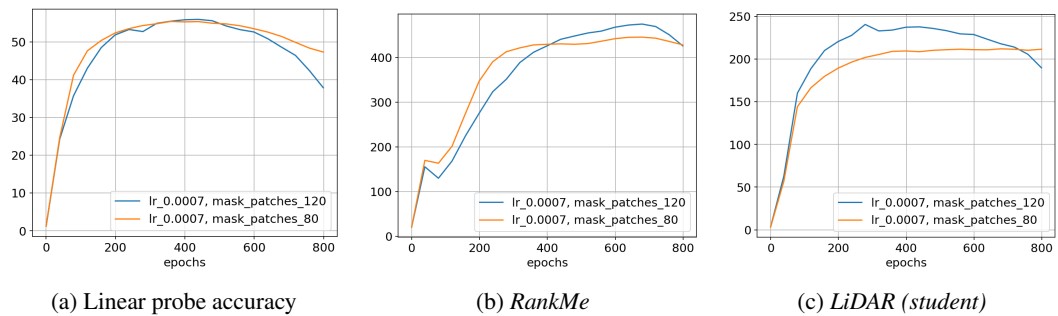

(a) Linear probe accuracy      (b) *RankMe*      (c) *LiDAR (student)*

Figure 16: data2vec: ViT-Base architecture trained on Imagenet-1K by varying the masking ratio. Plots show the evolution of metrics over training time.

| Hyperparameter | RankMe | RankMe (aug.) (Student) | LiDAR (Student) | RankMe (aug.) (Teacher) | LiDAR (Teacher) |
|---|---|---|---|---|---|
| Learning rate | 0.3639 | 0.4440 | **0.6680** | 0.4191 | 0.6240 |
| Mask ratio | 0.3720 | 0.3026 | **0.6089** | 0.2785 | 0.5466 |
| Overall | 0.3408 | 0.3795 | **0.7275** | 0.3627 | 0.6962 |

Table 14: data2vec: Spearman rank correlation coefficient for data2vec between effective ranks of RankMe, RankMe (aug.) and LiDAR and linear probe accuracy.

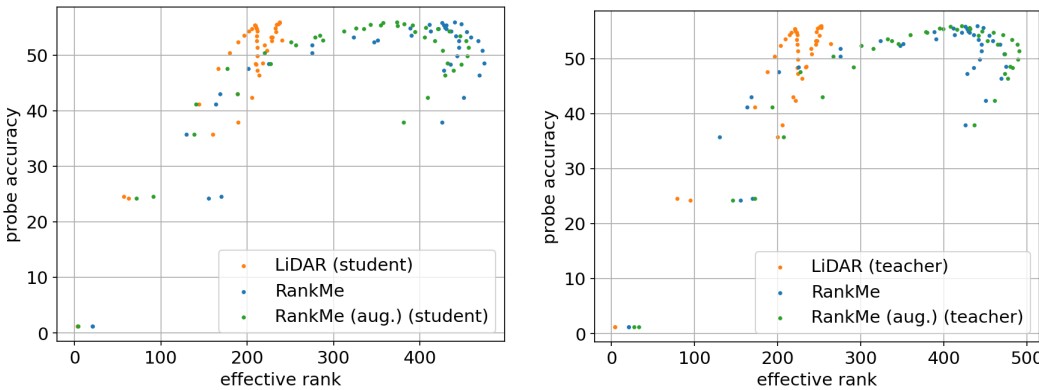

Figure 17: data2vec: Scatter plot of checkpoints collected during training of ViT-Base architecture on Imagenet-1K. The masking ratio was varied in this experiment.

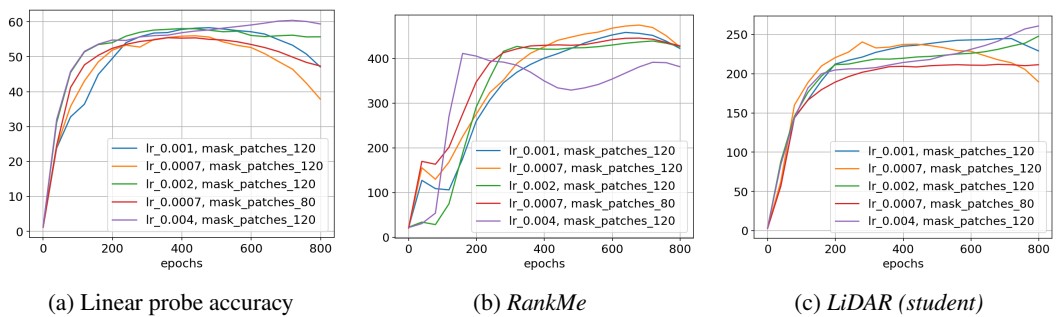

(a) Linear probe accuracy        (b) *RankMe*        (c) *LiDAR (student)*

Figure 18: data2vec: ViT-Base architecture trained on Imagenet-1K by varying one of learning rate or masking ratio. Plots show the evolution of metrics over training time.

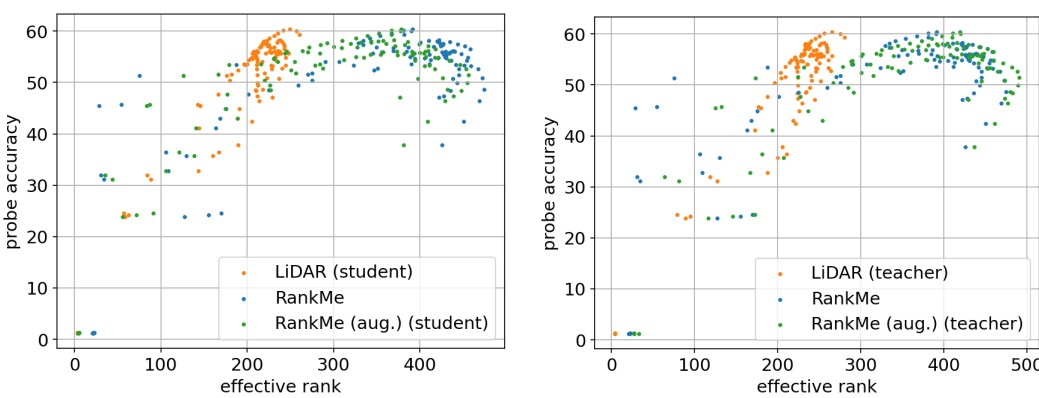

Figure 19: data2vec: Aggregated scatter plot of checkpoints collected during training of ViT-Base architecture on Imagenet-1K. Hyperparameters varied are one of learning rate and mask ratio (number of masked patches).

| Metric | LR | Mask ratio | Overall |
|--------|-----|-----------|---------|
| ImageNet Oracle | 60.3920 | 55.9780 | 60.3920 |
| RankMe | 48.6040 | 48.6980 | 48.6980 |
| RankMe (aug.) (Student) | 48.6040 | 51.4500 | 51.4500 |
| LiDAR (Student) | **59.3720** | **52.7460** | **59.3720** |
| RankMe (aug.) (Teacher) | 48.6040 | 51.4500 | 51.4500 |
| LiDAR (Teacher) | **59.3720** | **52.7460** | **59.3720** |

Table 15: data2vec: Linear probe accuracy recovered by *RankMe* and *LiDAR* on ImageNet-1K dataset.

## 9.4   DINO

DINO (Caron et al., 2021) is an example of a self-distillation approach to learning representations in a self-supervised manner. DINO uses a student and teacher encoder (weights updated via exponential moving average) ,multiple-crops and small patches to train a ViT (Dosovitskiy et al., 2021) in a self-supervised manner. Evaluations on multiple tasks presented in DINO (Caron et al., 2021) show that the resulting encoder learns strong representations. The loss function for DINO (Caron et al., 2021) minimizes the cross-entropy between the probability distributions provided by the teacher and student network. We reproduce a description below from DINO (Caron et al., 2021) for completeness:

$$\min_i H\left(P_t\left(z\right), P_s\left(z\right)\right) \tag{24}$$

where $H$ denotes the cross-entropy function, $P_s$ and $P_t$ denote the probability distributions output by the student and teacher networks respectively. The probabilities are computed via:

$$P_{net}(z)^{(i)} = \frac{\exp(f_{\theta_{net}}(z)^{(i)}/\tau_{net})}{\sum_{k=1}^{K}\exp(f_{\theta_{net}}(x)^{(k)}/\tau_{net})} \tag{25}$$

where the subscript "net" denotes either the student or the teacher network, $f$ is a neural network parameterized by $\theta$ and $z^{(i)}$ is a feature vector for sample $i$. A key hyperparameter in DINO (Caron et al., 2021) is the temperature value $\tau_s$ and $\tau_t$ used to control the sharpness of the output distribution produced by the softmax function. We test the performance of *RankMe*, augmented-*RankMe* and *LiDAR* to predict linear probing performance by varying these parameters. We use 1000 images from the source dataset and apply 50 augmentations used in DINO training to construct a dataset to calcuate *LiDAR*. We use 25600 samples to estimate *RankMe*.

We train a ViT-S with a patch size of $16 \times 16$ using the protocol described by DINO (Caron et al., 2021) and implemented in a reference implementation provided by the authors of DINO [7] on the Imagenet-1K (Russakovsky et al., 2015) dataset. The projection head consists of a 3-layer MLP with hidden dimension of 2048 and an output dimension referred to as bottleneck dimension of 256 that provides embeddings. These embeddings are that projected to a higher dimensional space of 65536 in our experiments. We use the embeddings to estimate *RankMe*, augmented-*RankMe* and *LiDAR* in our experiments consistent with the methodology adopted in *RankMe* (Garrido et al., 2022). We use an effective batch size of 512 images and train the model for 300 epochs. Our experiments consists of:

- varying the teacher temperature ($\tau_t$) while keeping the student temperature ($\tau_s$) fixed to 0.1. The values considered for ($\tau_t$) are $\{0.02, 0.04, 0.06, 0.07\}$
- varying the student temperature ($\tau_s$) while keeping the teacher temperature ($\tau_t$) fixed. The values considered for ($\tau_s$) are $\{0.2, 0.4\}$

We keep the rest of the training and linear probing hyperparameters identical to those provided in the official implementation [7]. The results of these experiments are available in Figure 20 and the correlation values are quantified in Table 16. Table 17 lists the linear probing accuracy recovered by ImageNet Oracle, *RankMe* and *LiDAR* metrics for the source dataset (ImageNet-1K).

---

[7]https://github.com/facebookresearch/dino

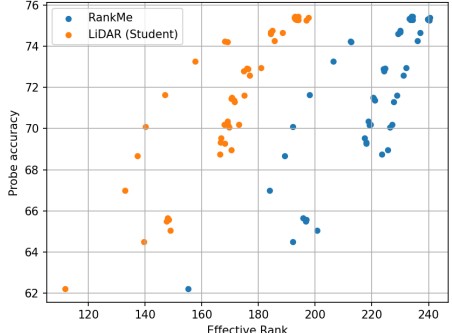 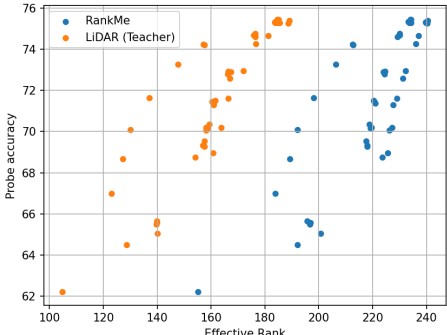

Figure 20: DINO: Aggregated scatter plot of checkpoints collected during training of ViT-Small architecture on Imagenet-1K. Hyperparameters varied are one of teacher or student softmax temperature described in the implementation section.

| Correlation | RankMe | LiDAR (Student) | LiDAR (Teacher) |
|---|---|---|---|
| Spearman rank | 0.8191 | 0.8807 | 0.8732 |
| Kendall's $\tau$ | 0.6288 | 0.7157 | 0.7299 |

Table 16: DINO: Compare *RankMe* and *LiDAR* with Spearman rank and Kendall's $\tau$ correlation coefficient metrics. Observe that *LiDAR* performs better than *RankMe* for both measures.

| Metric | Teacher temperature | Student temperature | Overall |
|---|---|---|---|
| ImageNet Oracle | 75.45 | 75.37 | 75.45 |
| RankMe | 75.38 | 75.28 | 75.38 |
| LiDAR (Student) | 75.38 | 75.28 | 75.38 |
| LiDAR (Teacher) | 75.38 | 75.28 | 75.38 |

Table 17: DINO: Average linear probe accuracy recovered by *RankMe* and *LiDAR* on ImageNet-1K. Hyperparameter values are described in Appendix 9.

## 9.5 SIMCLR

SimCLR (Chen et al., 2020) is an example of a contrastive joint-embedding self-supsersived learning method. The loss function for SimCLR is given by (Chen et al., 2020; Garrido et al., 2022) and included below for completeness:

$$L = - \sum_{i,j \in P} \frac{e^{Similarity(z_i, z_j)}}{\sum_{k=1}^{N} \mathbb{I}_{k \neq i} e^{Similarity(z_i, z_k)}} \tag{26}$$

where $Similarity$ denotes the cosine similarity between two vectors $z_i$ and $z_j$, $\mathbb{I}$ denotes the indicator function, $P$ is the set of all positive pairs and the number of examples in given by N.

We train a ResNet-50 backbone (He et al., 2016) with a 3 layer MLP projector with hidden dimensions 8192 and output dimension equal to 2048. In other words, the embeddings produced by SimCLR have a dimension equal to 2048. The network described above is trained with the LARS optimizer (You et al., 2017) and other settings described in (Chen et al., 2020) on the ImageNet-1k (Russakovsky et al., 2015) training split. The representations from the backbone are evaluated via standard linear probing by training a linear layer on ImageNet-1k training split and calculating test accuracy on the validation split. Probing is performed using SGD optimizer with Nesterov momentum with hyperparameters described in *RankMe* (Garrido et al., 2022). We use 5000 images from the source dataset (ImageNet-1K) and apply 10 augmentations per image to calculate *LiDAR*. We use 25600 images to calcuate *RankMe* to be consistent with the setup used by Garrido et al.

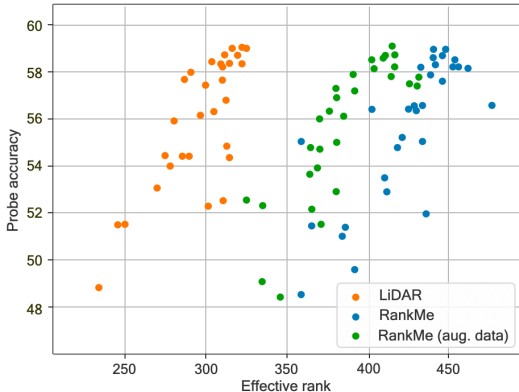

Figure 21: SimCLR: Performance of SimCLR representations measured by RankMe and LiDAR. Each point refers to a checkpoint evaluated with a **randomly searched** hyperparameter after 100 epochs of self-supervised pretraining. LiDAR shows strong correlation

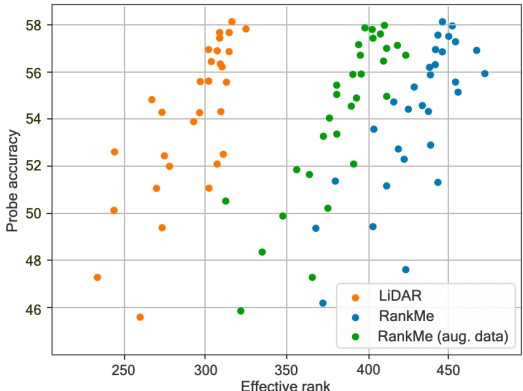

Figure 22: SimCLR: Performance of SimCLR representations measured by RankMe and LiDAR. Each point refers to a checkpoint evaluated with the **grid search** hyperparameter from Rankme paper after 100 epochs of self-supervised pretraining. LiDAR shows strong correlation

(2022). We consider two sets of experiments to test the performance of *LiDAR* and *RankMe* with SimCLR (Chen et al., 2020):

- A grid search with the hyperparametrs sets that include learning rate, weight decay, embedding dimension and softmax temperature used in *RankMe* (Garrido et al., 2022). Figure 22 shows a scatter plot of the probe accuracy vs. effective rank estiamted by *RankMe*, augmented-*RankMe* and *LiDAR*. Table 19 quantifies the correlations for the above methods. Table 21 shows the linear probe accuracy recovered by the various metrics and compares the results to ImageNet Oracle

- A random hyperparameter search by randomly sampling learning rate, weight decay and softmax temperature to generate hyperparameter sets. We create a table of hyperpameters with learning rate chosen from $[0.3, 0.4, 0.5, 0.6]$, weight decay chosen from $\left[10^{-7}, 10^{-6}, 10^{-5}\right]$ and softmax temperature chosen from $[0.05, 0.1, 0.15, 0.2, 0.25]$. We select 30 sets of hyperparameters from this table and use these to train models with Sim-CLR and evaluate performance. Figure 21 shows the results for this experiment as a scatter plot and Table 19 quantifies the correlations.

| Hyperparameter | RankMe | RankMe (aug. dataset) | LiDAR |
|---|---|---|---|
| Learning rate | 0.5248 | 0.5301 | **0.9138** |
| Weight decay | 0.4019 | 0.5573 | **0.8964** |
| Temperature | 0.5704 | 0.6764 | **0.9182** |
| Overall | 0.5125 | 0.6304 | **0.9155** |

(a) Spearman Rank correlation coefficient

| Hyperparameter | RankMe | RankMe (aug. dataset) | LiDAR |
|---|---|---|---|
| Learning rate | 0.5706 | 0.4694 | **0.7512** |
| Weight decay | 0.3580 | 0.4221 | **0.7148** |
| Temperature | 0.4209 | 0.7061 | **0.8435** |
| Overall | 0.4906 | 0.5581 | **0.7967** |

(b) Kendall's $\tau$ correlation coefficient

Table 18: SimCLR: Correlation between effective rank estiamted by *RankMe* and *LiDAR* and probe accuracy per hyperparameter.

| Correlation | RankMe | RankMe (aug. dataset) | LiDAR |
|---|---|---|---|
| Spearman rank | 0.5125 | 0.6304 | **0.9155** |
| Kendall's $\tau$ | 0.4906 | 0.5581 | **0.7967** |

Table 19: SimCLR: Compare RankMe and LiDAR using Spearman Rank correlation and Kendall's $\tau$ correlation measures evaluated during hyperparameter **grid search**.

| Correlation | RankMe | RankMe (aug. dataset) | LiDAR |
|---|---|---|---|
| Spearman rank | 0.5301 | 0.6389 | **0.9188** |
| Kendall's $\tau$ | 0.4982 | 0.5761 | **0.8167** |

Table 20: SimCLR: Compare RankMe and LiDAR using Spearman Rank correlation and Kendall's $\tau$ correlation measures evaluated during hyperparameter **random search**.

| Metric | LR | WD | Temp. | Overall |
|---|---|---|---|---|
| Imagenet Oracle | 58.2370 | 56.6740 | 57.4710 | 59.1420 |
| *RankMe* | 55.8950 | 55.1490 | 56.0390 | 56.4630 |
| *RankMe* (aug. dataset) | 57.2010 | 55.8170 | 56.3170 | 57.8260 |
| *LiDAR* | **57.8940** | **56.3020** | **57.0920** | **58.9270** |

Table 21: SimCLR: Linear probe accuracy recovered by *RankMe*, augmented-*RankMe* and *LiDAR* on ImageNet-1K dataset at the end of training. Results are from trials run with hyperparameter **random search**.

| Metric | LR | WD | Target mask scale | Context mask scale | Overall |
|---|---|---|---|---|---|
| ImageNet Oracle | 78.49 | 78.49 | 78.49 | 78.49 | 78.49 |
| RankMe | 72.09 | 75.45 | 75.45 | 75.15 | 72.09 |
| RankMe (aug.) (Student) | **77.78** | **77.78** | **77.78** | 77.78 | **77.78** |
| RankMe (aug.) (Teacher) | 72.09 | 72.14 | 70.20 | 75.15 | 72.09 |
| LiDAR (Student) | **77.78** | **77.78** | **77.78** | 77.78 | **77.78** |
| LiDAR (Teacher) | 74.33 | 76.84 | 73.91 | **78.30** | 74.33 |

Table 22: I-JEPA: Average linear probe accuracy recovered by *RankMe* and *LiDAR* on OOD datasets. Hyperparameters set via grid search.

## 10 OUT-OF-DISTRIBUTION (OOD) DATASETS EVALUATION

### 10.1 EVALUATION PROTOCOL

In order to evaluate *RankMe* and *LiDAR* methods, we use CIFAR10, CIFAR100 (Krizhevsky & Hinton, 2009), EuroSAT (Helber et al., 2019), Food101 (Bossard et al., 2014) and SUN397 (Xiao et al., 2010) as unseen or OOD datasets. These commonly used datasets are different from our source dataset, ImageNet-1k, and are used to evaluate the effectiveness of representations on downstream tasks. The downstream task protocol considered here is linear probing with a frozen backbone. We follow the protocol used in VISSL (Goyal et al., 2021) to train and evaluate a linear probe with the above OOD datasets.

For all datasets, we use stochastic gradient descent with Nesterov (SGD w/Nesterov) momentum to optimize the linear classification layer. We use a learning rate of $0.01$, momentum value of $0.9$, weight decay $0.0005$ and cross-entropy loss to train the linear layer for 28 epochs. We use a multi-step learning rate schedule where the learning rate is dropped by a factor of $0.1$ every 8 steps. We follow the data augmentation procedure used in VISSL (Goyal et al., 2021) that consists of random resized crops and random horizontal flips for training and center crop for evaluation. The results of the OOD datasets evaluation are listed in the following sections.

### 10.2 I-JEPA

We study the OOD performance of I-JEPA model pretrained on ImageNet-1K as described in Appendix 9.2. The checkpoints chosen by ImageNet Oracle, *RankMe*, its augmented variant and *LiDAR* are trained on 5 OOD datasets using the protocol described in Appendix 10.1. Table 22 and Table 24 lists the average probe accuracy calculated over the five OOD datasets for hyperparameters selected by grid search and random search, respectively. We observe from Table 22 that *LiDAR* calculated with student branch outperforms *RankMe* in terms of linear probe accuracy on OOD datasets except for the context scale parameter where-in *LiDAR* calculated with the teacher branch performs slightly better than its student counterpart. This observation is an interesting contrast to the corresponding in-domain linear probe accuracy shown in Table 3 where-in the student branch-based *LiDAR* works well across all hyperparameters. Another remarkable observation here is that our proposed augmented-*RankMe* variant performs better than vanilla *RankMe* on OOD dataset which is consistent with our observations made with in-domain data in Table 3. Finally, we see from Table 22 that while ImageNet Oracle shows the best performance on OOD datasets, *LiDAR* is able to recover most of the linear probe accuracy on these datasets as well. Table 23 lists a per-dataset breakdown of the linear probe accuracy for checkpoints selected by various metrics.

Table 24 shows the performance of I-JEPA checkpoints on OOD datasets trained with hyperparameters selected via random search. We observe from Table 24 that both variants of *LiDAR* outperform *RankMe* and its augmented variants and remarkably outperforms the OOD linear probe accuracy recovered by the ImageNet Oracle. This observation shows that OOD performance can be improved under certain conditions which is also an observation made by Garrido et al. (2022). Table 25 lists a per-dataset breakdown of the linear probe accuracy for checkpoints selected by various metrics.

| Dataset | Metric | Hyperparameters | | | | |
|---|---|---|---|---|---|---|
| | | LR | WD | Target mask scale | Context mask scale | Overall |
| CIFAR10 | ImageNet Oracle | 89.66 | 89.66 | 89.66 | 89.66 | 89.66 |
| | RankMe | 86.92 | 88.8 | 88.8 | 87.75 | 86.92 |
| | RankMe (aug.) (Student) | 89.64 | 89.64 | 89.64 | 89.64 | 89.64 |
| | RankMe (aug.) (Teacher) | 86.92 | 86.04 | 85.04 | 87.75 | 86.92 |
| | LiDAR (Student) | 89.64 | 89.64 | 89.64 | 89.64 | 89.64 |
| | LiDAR (Teacher) | 87.86 | 89.29 | 82.73 | 89.99 | 87.66 |
| CIFAR100 | ImageNet Oracle | 69.03 | 69.03 | 69.03 | 69.03 | 69.03 |
| | RankMe | 63.57 | 67.19 | 67.19 | 66.52 | 63.57 |
| | RankMe (aug.) (Student) | 68.81 | 68.81 | 68.81 | 68.81 | 68.81 |
| | RankMe (aug.) (Teacher) | 63.57 | 63.97 | 63.07 | 66.52 | 63.57 |
| | LiDAR (Student) | 68.81 | 68.81 | 68.81 | 68.81 | 68.81 |
| | LiDAR (Teacher) | 65.04 | 68.66 | 56.46 | 69.28 | 65.04 |
| EuroSAT | ImageNet Oracle | 95.02 | 95.02 | 95.02 | 95.02 | 95.02 |
| | RankMe | 94.56 | 95.68 | 95.68 | 95.62 | 94.56 |
| | RankMe (aug.) (Student) | 95.18 | 95.18 | 95.18 | 95.18 | 95.18 |
| | RankMe (aug.) (Teacher) | 94.56 | 95.42 | 94.92 | 95.62 | 94.56 |
| | LiDAR (Student) | 95.18 | 95.18 | 95.18 | 95.18 | 95.18 |
| | LiDAR (Teacher) | 93.16 | 95.54 | 94.6 | 95.6 | 93.16 |
| Food101 | ImageNet Oracle | 72.65 | 72.65 | 72.65 | 72.65 | 72.65 |
| | RankMe | 58.93 | 65.22 | 65.22 | 65.49 | 58.93 |
| | RankMe (aug.) (Student) | 70.49 | 70.49 | 70.49 | 70.49 | 70.49 |
| | RankMe (aug.) (Teacher) | 58.93 | 59.49 | 55.78 | 65.49 | 58.93 |
| | LiDAR (Student) | 70.79 | 70.79 | 70.79 | 70.79 | 70.79 |
| | LiDAR (Teacher) | 65.13 | 67.96 | 71.59 | 71.86 | 65.13 |
| SUN397 | ImageNet Oracle | 66.08 | 66.08 | 66.08 | 66.08 | 66.08 |
| | RankMe | 56.46 | 60.36 | 60.36 | 60.37 | 56.46 |
| | RankMe (aug.) (Student) | 64.47 | 64.47 | 64.47 | 64.47 | 64.47 |
| | RankMe (aug.) (Teacher) | 56.46 | 55.8 | 52.17 | 60.37 | 56.46 |
| | LiDAR (Student) | 64.47 | 64.47 | 64.47 | 64.47 | 64.47 |
| | LiDAR (Teacher) | 60.47 | 62.75 | 64.16 | 64.78 | 60.47 |

Table 23: I-JEPA: Top-1 accuracies computed on representations when tuning hyperparameters with ImageNet validation performance, *RankMe*, augmented *RankMe* and *LiDAR*. Hyperparameters set via grid search.

| ImageNet Oracle | RankMe | RankMe (aug.) (Student) | RankMe (aug.) (Teacher) | LiDAR (Student) | LiDAR (Teacher) |
|---|---|---|---|---|---|
| 74.85 | 73.24 | 73.24 | 73.24 | 76.23 | **77.34** |

Table 24: I-JEPA: Average linear probe accuracy recovered by *RankMe* and *LiDAR* on OOD datasets. Hyperparameters set via random search.

| Dataset | ImageNet Oracle | RankMe | RankMe (aug.) (Student) | RankMe (aug.) (Teacher) | LiDAR (Student) | LiDAR (Teacher) |
|---|---|---|---|---|---|---|
| CIFAR10 | 82.69 | 87.32 | 87.32 | 87.32 | 86.21 | 87.29 |
| CIFAR100 | 58.66 | 65.98 | 65.98 | 65.98 | 65.92 | 64.91 |
| EuroSAT | 95.60 | 95.68 | 95.68 | 95.68 | 96.04 | 96.06 |
| Food101 | 70.94 | 61.16 | 61.16 | 61.16 | 69.60 | 71.86 |
| SUN397 | 66.36 | 56.04 | 56.04 | 56.04 | 63.38 | 66.59 |

Table 25: I-JEPA: Top-1 accuracies computed on representations when tuning hyperparameters with ImageNet validation performance, *RankMe*, augmented *RankMe* and *LiDAR*. Hyperparameters set via random search

| Metric | LR | Mask ratio | Overall |
|---|---|---|---|
| ImageNet Oracle | 68.69 | 69.07 | 68.69 |
| RankMe | 58.86 | 58.86 | 58.86 |
| RankMe (aug.) (Student) | 58.86 | 60.76 | 60.76 |
| RankMe (aug.) (Teacher) | 58.86 | 60.76 | 60.76 |
| LiDAR (Student) | **63.76** | **73.07** | **63.76** |
| LiDAR (Teacher) | **63.76** | **73.07** | **63.76** |

Table 26: data2vec: Average linear probe accuracy recovered by *RankMe* and *LiDAR* on OOD datasets. Hyperparameters set via grid search.

## 10.3 DATA2VEC

In this section, we study the OOD performance of data2vec models trained on ImageNet-1K the details of which are described in Appendix 9.3. We select checkpoints based on ImageNet Oracle, *RankMe* and *LiDAR* as described in Appendix 9.3 and train them on the 5 OOD datasets described in Appendix 10.1. Table 26 shows the average linear probe accuracy of data2vec checkpoints chosen by ImageNet Oracle, *RankMe*, augmented-*RankMe* and *LiDAR*. We observe from Table 26 that *LiDAR* outperforms *RankMe* and its augmented variant and in the case of mask ratio hyperparameter outperforms the ImageNet Oracle as well. The latter observation supports those made by Garrido et al. (2022) on the need to have new methods that do not rely on ImageNet performance alone for checkpoint selection. Table 27 lists the per-dataset breakdown of linear probe accuracy for data2vec checkpoints chosen by various metrics listed above.

## 10.4 DINO

In this section, we study the OOD performance of DINO models trained on ImageNet-1K the details of which are described in Appendix 9.4. We select checkpoints based on ImageNet Oracle, *RankMe* and *LiDAR* as described in Appendix 9.4 and train them on the 5 OOD datasets described in Appendix 10.1. Table 28 lists the average linear probe accuracy for the OOD datasets from which we observe that *LiDAR* is able to show the same as or slightly better performance than both *RankMe* and ImageNet Oracle for DINO hyperparameter selection. Table 29 shows a per-dataset breakdown of probe accuracy on OOD datasets.

## 10.5 VICREG

The VICReg model is pretrained on ImageNet using the same hyperparameter grid as the one published in (Garrido et al., 2022) (each grid point for 100 epochs). From the set of the last checkpoints per grid point, a pool of the best-performing models, according to ImageNet validation performance, *RankMe*, and the proposed *LiDAR*, was selected. We conducted probe analysis on the resulting pool of checkpoints using five different OOD datasets. The probe analysis employed learning parameters reported in (Garrido et al., 2022) (standard protocol in VISSL (Goyal et al., 2021)). Table 30 below summarizes the probe accuracies averaged over the 5 OOD datasets considered in our evaluation for the checkpoint selected by each metric (one of ImageNet Oracle, *RankMe* and *LiDAR*) for a given hyperparameter. We observe that *LiDAR* consistently outperforms *RankMe* across all hyperparam-

| Dataset | Metric | Hyperparameters | | |
| --- | --- | --- | --- | --- |
| | | LR | Mask ratio | Overall |
| CIFAR10 | ImageNet Oracle | 71.63 | 74.62 | 71.63 |
| | RankMe | 60.21 | 60.21 | 60.21 |
| | RankMe (aug.) (Student) | 60.21 | 61.11 | 61.11 |
| | RankMe (aug.) (Teacher) | 60.21 | 61.11 | 61.11 |
| | LiDAR (Student) | 63.87 | 84.46 | 63.87 |
| | LiDAR (Teacher) | 63.87 | 84.46 | 63.87 |
| CIFAR100 | ImageNet Oracle | 42.48 | 46.85 | 42.28 |
| | RankMe | 31.15 | 31.15 | 31.15 |
| | RankMe (aug.) (Student) | 31.15 | 31.88 | 31.88 |
| | RankMe (aug.) (Teacher) | 31.1 | 31.88 | 31.88 |
| | LiDAR (Student) | 33.98 | 60.20 | 33.98 |
| | LiDAR (Teacher) | 33.98 | 60.20 | 33.98 |
| EuroSAT | ImageNet Oracle | 93.06 | 92.98 | 93.06 |
| | RankMe | 84.30 | 84.30 | 84.30 |
| | RankMe (aug.) (Student) | 84.30 | 85.22 | 85.22 |
| | RankMe (aug.) (Teacher) | 84.30 | 85.22 | 85.22 |
| | LiDAR (Student) | 87.20 | 95.12 | 87.20 |
| | LiDAR (Teacher) | 87.20 | 95.12 | 87.20 |
| Food101 | ImageNet Oracle | 70.41 | 68.16 | 70.41 |
| | RankMe | 61.18 | 61.18 | 61.18 |
| | RankMe (aug.) (Student) | 61.18 | 65.64 | 65.64 |
| | RankMe (aug.) (Teacher) | 61.18 | 65.64 | 65.64 |
| | LiDAR (Student) | 69.04 | 64.80 | 69.04 |
| | LiDAR (Teacher) | 69.04 | 64.80 | 69.04 |
| SUN397 | ImageNet Oracle | 66.09 | 62.73 | 66.09 |
| | RankMe | 57.47 | 57.47 | 57.47 |
| | RankMe (aug.) (Student) | 57.47 | 59.98 | 59.98 |
| | RankMe (aug.) (Teacher) | 57.47 | 59.98 | 59.98 |
| | LiDAR (Student) | 64.72 | 60.79 | 64.72 |
| | LiDAR (Teacher) | 64.72 | 60.79 | 64.72 |

Table 27: data2vec: Top-1 accuracies computed on representations when tuning hyperparameters with ImageNet validation performance, *RankMe*, augmented *RankMe* and *LiDAR*. Hyperparameters set via grid search.

| Metric | Teacher temperature | Student temperature | Overall |
| --- | --- | --- | --- |
| ImageNet Oracle | 84.66 | 84.59 | 84.66 |
| RankMe | 84.66 | 84.46 | 84.66 |
| LiDAR (Student) | 84.66 | 84.72 | 84.66 |
| LiDAR (Teacher) | 84.66 | 84.72 | 84.66 |

Table 28: DINO: Average linear probe accuracy recovered by *RankMe* and *LiDAR* on OOD datasets. Hyperparameter values are described in Appendix 9.

| Dataset | Metric | Hyperparameters | | |
| --- | --- | --- | --- | --- |
| | | Teacher temperature | Student temperature | Overall |
| CIFAR10 | ImageNet Oracle | 95.41 | 95.35 | 95.41 |
| | RankMe | 95.40 | 95.32 | 95.40 |
| | LiDAR (Student) | 95.40 | 95.51 | 95.40 |
| | LiDAR (Teacher) | 95.40 | 95.51 | 95.40 |
| CIFAR100 | ImageNet Oracle | 81.27 | 81.31 | 81.27 |
| | RankMe | 81.67 | 81.31 | 81.67 |
| | LiDAR (Student) | 81.67 | 81.94 | 81.67 |
| | LiDAR (Teacher) | 81.67 | 81.94 | 81.67 |
| EuroSAT | ImageNet Oracle | 96.38 | 96.40 | 96.38 |
| | RankMe | 96.68 | 96.30 | 96.68 |
| | LiDAR (Student) | 96.68 | 96.30 | 96.68 |
| | LiDAR (Teacher) | 96.68 | 96.30 | 96.68 |
| Food101 | ImageNet Oracle | 79.23 | 79.19 | 79.23 |
| | RankMe | 79.25 | 78.99 | 79.25 |
| | LiDAR (Student) | 79.25 | 79.47 | 79.25 |
| | LiDAR (Teacher) | 79.25 | 79.47 | 79.25 |
| SUN397 | ImageNet Oracle | 71.00 | 70.70 | 71.00 |
| | RankMe | 70.29 | 70.40 | 70.29 |
| | LiDAR (Student) | 70.29 | 70.39 | 70.29 |
| | LiDAR (Teacher) | 70.29 | 70.39 | 70.29 |

Table 29: DINO: Top-1 accuracies computed on representations when tuning hyperparameters with ImageNet validation performance, *RankMe*, augmented *RankMe* and *LiDAR*. Hyperparameter values are described in Appendix 9.

| Method | Cov | inv | LR | WD |
|---|---|---|---|---|
| ImageNet Oracle | 77.83 | 77.61 | 76.94 | 76.97 |
| RankMe | 77.53 | 75.53 | 76.94 | 74.93 |
| LiDAR | 78.04 | 77.74 | 76.94 | 74.93 |

Table 30: VICReg: Average Top-1 probe accuracies (across OOD datasets) computed on representation (following VISSL protocol) when tuning hyperparameters with ImageNet validation performance.

| Dataset | Method | Cov | inv | LR | WD |
|---|---|---|---|---|---|
| CIFAR10 | ImageNet Oracle | 88.49 | 88.06 | 87.25 | 87.48 |
| | RankMe | 88.22 | 87.05 | 87.25 | 86.54 |
| | LiDAR | 88.68 | 88.47 | 87.25 | 86.54 |
| CIFAR100 | ImageNet Oracle | 69.62 | 69.46 | 68.76 | 69.5 |
| | RankMe | 69.93 | 67.83 | 68.76 | 67.02 |
| | LiDAR | 70.65 | 70.2 | 68.76 | 67.02 |
| EuroSAT | ImageNet Oracle | 95.9 | 95.88 | 95.0 | 95.6 |
| | RankMe | 95.46 | 94.64 | 95.0 | 94.88 |
| | LiDAR | 95.94 | 95.66 | 95.0 | 94.88 |
| Food101 | ImageNet Oracle | 70.07 | 69.66 | 69.37 | 68.62 |
| | RankMe | 69.43 | 66.34 | 69.37 | 65.35 |
| | LiDAR | 70.01 | 69.84 | 69.37 | 65.35 |
| SUN397 | ImageNet Oracle | 65.09 | 64.99 | 64.32 | 63.66 |
| | RankMe | 64.63 | 61.79 | 64.32 | 60.86 |
| | LiDAR | 64.94 | 64.51 | 64.32 | 60.86 |

Table 31: VICReg: Top-1 probe accuracies computed on representation (following VISSL protocol) when tuning hyperparameters with ImageNet validation performance.

eters except learning rate where *LiDAR* matches *RankMe*. Additionally, we observe that *LiDAR* is able to select hyperparameters that improve the performance over ImageNet Oracle in certain cases which adds to the observations made by Garrido et al. (2022) on the need to have new methods to select hyperparameters. Table 31 provides a per-dataset breakdown of the linear probe accuracy on the 5 downstream datasets considered in our expeirments.

## 10.6 SIMCLR

The SimCLR model is pretrained on ImageNet using the random hyperparameter search policy as specified in Appendix 9.5. For better evaluation performance on OOD datasets, we use stronger augmentations during SimCLR pretraining. Specifically, besides the default augmentations, we follow the original SimCLR paper (Chen et al., 2020) to use extra augmentations of Sobel filtering, equalization and motion blur. We conducted probe analysis on five OOD datasets using the learning parameters listed in (Garrido et al., 2022). Table 32 below summarizes the average probe accuracies for the checkpoint selected by a given metric (ImageNet validation performance, RankMe, and our LiDAR) per hyperparameter. We observe that *LiDAR* outperforms *RankMe* in being able to select better hyperparameters across all settings considered in our experiments. *LiDAR* also recovers most of the performance seen with ImageNet Oracle. Table 33 provides a per-dataset breakdown of the three methods used to select hyperparameters.

| Method | Temp | LR | WD |
|---|---|---|---|
| ImageNet Oracle | 80.68 | 80.84 | 80.76 |
| RankMe | 78.10 | 79.48 | 79.36 |
| LiDAR | 79.88 | 80.30 | 80.24 |

Table 32: SimCLR: Average Top-1 probe accuracies (across OOD datasets) computed on representation when tuning hyperparameters (random search) based on ImageNet validation performance.

| Dataset | Method | Temp | LR | WD |
|---|---|---|---|---|
| CIFAR10 | ImageNet Oracle | 90.51 | 90.83 | 90.07 |
| | RankMe | 88.24 | 89.57 | 90.14 |
| | LiDAR | 89.78 | 90.13 | 90.54 |
| CIFAR100 | ImageNet Oracle | 73.52 | 73.47 | 73.38 |
| | RankMe | 70.36 | 72.71 | 73.32 |
| | LiDAR | 72.63 | 73.18 | 73.43 |
| EuroSAT | ImageNet Oracle | 96.27 | 96.51 | 96.54 |
| | RankMe | 95.24 | 95.68 | 95.73 |
| | LiDAR | 95.92 | 96.31 | 95.39 |
| Food101 | ImageNet Oracle | 74.63 | 64.79 | 74.84 |
| | RankMe | 71.53 | 72.58 | 71.22 |
| | LiDAR | 73.78 | 74.13 | 73.84 |
| SUN397 | ImageNet Oracle | 68.47 | 68.59 | 69.03 |
| | RankMe | 65.23 | 66.84 | 66.46 |
| | LiDAR | 67.34 | 67.83 | 68.06 |

Table 33: SimCLR: Top-1 probe accuracies computed on representation when tuning hyperparameters (random search) based on ImageNet validation performance.

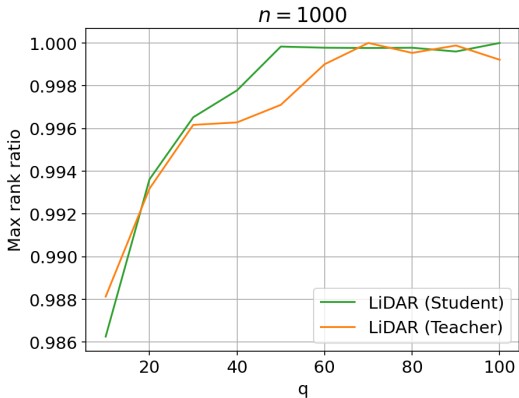

Figure 23: *LiDAR* sensitivity to the number of samples per class $q$ hyperparameter. $q$ is varied from 10 to 100 while $n$ is fixed to 1000 to evaluate an I-JEPA checkpoint.

## 11    SENSITIVITY OF LIDAR HYPERPARAMETERS

We note from Section 4.3 that the rank of *LiDAR* is bounded by the number of surrogate classes $n$ which in our case corresponds to the number of "clean" image samples we draw at random from the source dataset. In practice we recommend choosing a value for $n$ that is greater than the length of feature vectors ($p$) (representations or emebddings) to avoid imposing an artificial constraint on the ranks estimated by *LiDAR*. The number of samples per class $q$ is another important hyperparameter. We conduct an ablation to study the impact of this hyperparameter with I-JEPA embeddings. Figure 23 shows the results from this ablation. As we can see from Figure 23, *LiDAR* for both the teacher and student is already close to 99% of its final value at $q = 10$ and reaches more than 99% of its final value at $q = 50$, which is the value used in our experiments.

### 11.1    HYPERPARAMETER SELECTION

The results from Appendix 11 suggest that a value of $n = 1000$ and $q = 50$ provides a satisfactory estimate of *LiDAR* for I-JEPA. We copy these hyperparameters to data2vec and DINO settings as these methods produce features that are smaller than or equal to the features provided by I-JEPA. Our empirical results with these methods shown in the main text and the appendix indicate that these hyperparameters are sufficient to saturate performance.

SimCLR and VICReg produce features of length 2048 that requires us to use a higher value of $n = 5000$ to ensure good estimates of the effective rank. We lower the value of $q$ to 10 to ensure that the size of *LiDAR* dataset is the same across all of our evaluations.

| Method | Sample count | Median time (std. dev.) (seconds) | Feature extraction (seconds) |
|---|---|---|---|
| RankMe | 10K | 0.38 (2.38) | 11.29 |
| LiDAR (Student) | 10K (n=1K, q=10) | 0.53 (0.07) | 23.22 |
| LiDAR (Student) | 50K (n=1K, q=50) | 0.71 (0.06) | 96.95 |

Table 34: Wall-clock time to calculate effective rank for *RankMe* and *LiDAR*. The emebddings are gathered from an I-JEPA (Assran et al., 2023) checkpoint for all methods.

## 12  RUNTIME COMPARISON OF *LiDAR* AND *RankMe*

Table 34 shows the wall-clock time of *RankMe* and *LiDAR* methods. We use the checkpoint that corresponds to the ImageNet Oracle for I-JEPA and calculate *RankMe* and *LiDAR* using the above checkpoint. The feature extraction uses a batch size of 200 and uses $10K$ samples for *RankMe* and $10K$ for *LiDAR* to ensure comparison is conducted with the same number of samples and $50K$ as this is the operating point for *LiDAR* with I-JEPA. The feature extraction is done on one GPU while the metrics are implemented on the CPU. We run 10 trials to ensure we can obtain a reasonable estimate of run-time variability. We present the median wall-clock times in Table 34 to guard against outliers while we also show the standard deviation in parentheses. We observe from Table 34 that *RankMe* is faster than *LiDAR* as expected. However, both methods are significantly faster than the feature extraction step, pinpointing the main place where optimizing the pipeline can be beneficial. We note here that feature extraction for *LiDAR* involves running multiple forward passes through a predictor network for I-JEPA (Assran et al., 2023) which adds to the compute time. In this work, we use standard off-the-shelf implementations for *RankMe* and *LiDAR*. A detailed study of an efficient implementation for *LiDAR* and its complexity analysis is left as a topic for future work. The work in (Cai et al., 2008) could be a relevant starting point. Notably, the running time difference between these methods is of the order of seconds/minutes which is negligible compared to model training or downstream task probing that may require additional data and/or hyperparameter optimization.

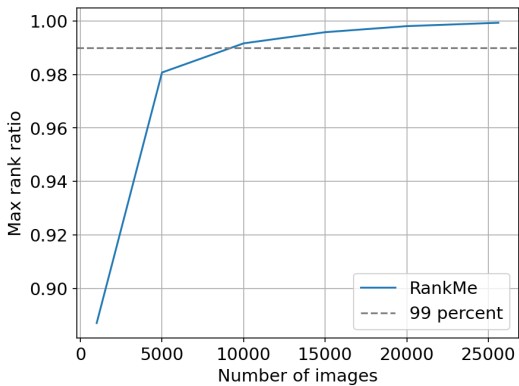

Figure 24: *RankMe* estimates versus the number of samples for I-JEPA.

## 13  *RankMe* ESTIMATE VERSUS NUMBER OF INPUT SAMPLES

**I-JEPA:**   We run an experiment with an I-JEPA (Assran et al., 2023) checkpoint to determine the beahvior of *RankMe* estimates as a function of the number of samples. We vary the number of samples from $1000$ to $25600$ where the former number corresponds to the number of samples $n$ used in *LiDAR* while $25600$ samples represents the number of samples recommended by Garrido et al. (2022). We note here that the value of $25600$ is determined for $2048$-dimensional vectors while I-JEPA outputs $768$-dimensional vectors. We observe from Figrue 24 that we obtain a rank estimate that is greater than $99\%$ of its maximum value by just using $10000$ samples. This ablation suggests that the number of samples used to estimate *RankMe* is reasonable for I-JEPA and data2vec that produce feature vectors that have identical sizes.

## 14 NUMBER OF SAMPLES FOR *RankMe* WITH I-JEPA AND DATA2VEC REPRESENTATIONS

| Hyperparameter | 10000 samples (RankMe) | 25600 samples (RankMe) |
|---|---|---|
| Learning rate | 0.6835 | 0.6835 |
| Weight decay | 0.5040 | 0.5050 |
| Target mask scale | 0.2867 | 0.2847 |
| Context mask scale | 0.4246 | 0.4256 |
| Overall | 0.5830 | 0.5829 |

Table 35: I-JEPA: Kendall's $\tau$ coefficient between effective ranks of *RankMe* and linear probe accuracy. *RankMe* estimated with 10000 samples and 25600 samples. Hyperparameters are varied via grid sampling.

| Hyperparameter | 10000 samples (RankMe) | 25600 samples (RankMe) |
|---|---|---|
| Overall | 0.8314 | 0.8308 |

Table 36: I-JEPA: Kendall's $\tau$ coefficient between effective ranks of *RankMe* and linear probe accuracy. *RankMe* estimated with 10000 samples and 25600 samples. Hyperparameters are varied via random sampling.

| Hyperparameter | 10000 samples (RankMe) | 25600 samples (RankMe) |
|---|---|---|
| Learning rate | 0.2410 | 0.2392 |
| Mask ratio | 0.2683 | 0.2706 |
| Overall | 0.2238 | 0.2216 |

Table 37: data2vec: Kendall's $\tau$ coefficient between effective ranks of *RankMe* and linear probe accuracy. *RankMe* estimated with 10000 samples and 25600 samples. Hyperparameters are varied via grid sampling.

Our analysis in Appendix 13 suggests that using 10000 samples may be sufficient to obtain a highly quality of estimate of the effective rank of the embeddings, i.e., *RankMe*. Garrido et al. (2022) suggest using 25600 samples based on their analysis of with 2048-dimensional vectors. We study the impact of the number of samples on *RankMe* as it pertains to the correlation between *RankMe* and linear probing performance. We test the behavior of *RankMe* with 10000 and 25600 samples and calculate the correlation between *RankMe* linear probe accuracy. Table 35, Table 36 and Table 37 show the results of this analysis for I-JEPA with hyperparameters set via grid search, I-JEPA wtih hyperparameters set via random search and data2vec with hyperparameters set via grid search respectively. The Kendall's $\tau$ correlation coefficient shows minor differences and the linear probe accuracy recovered for these models for in-domain data is identical (Table 3, Table 4, Table 6) and the performance on OOD data is also identical (accuracies available in Appendix 10) as both estimates of *RankMe* recover identical checkpoints. Figure 25, Figure 26, and Figure 27, show a scatter plot of linear probe accruacy versus *RankMe*. We observe that while there are minor differences in the effective ranks calculated with the number of input samples, the overall shape of the distribution that is also captured by Kendall's-$\tau$ are very similar which is also reflected in the data presented in the tables above. This ablation suggests that the predictive performance of *RankMe* on I-JEPA and data2vec representations is stable once with a sufficiently large number of samples.

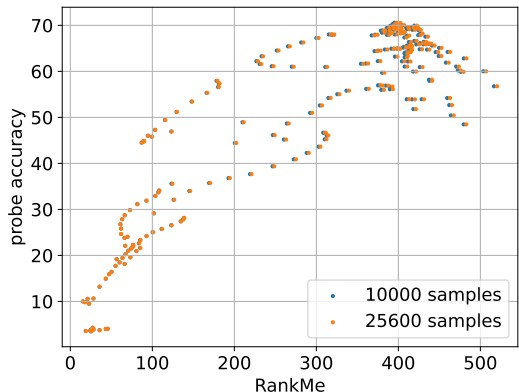

Figure 25: I-JEPA: Effect of the number of samples on *RankMe*. Parameters set via grid search.

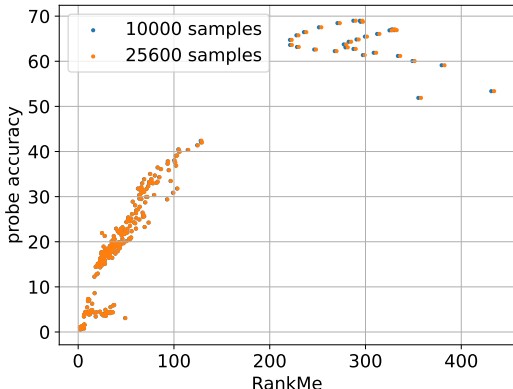

Figure 26: I-JEPA: Effect of the number of samples on *RankMe*. Parameters set via random search.

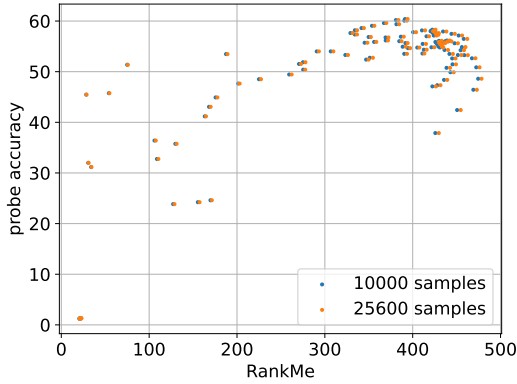

Figure 27: data2vec: Effect of the number of samples on *RankMe*. Parameters set via grid search.

