# OpenReview forum: "LiDAR: Sensing Linear Probing Performance in Joint Embedding SSL Architectures"
_ICLR.cc/2024/Conference — ICLR 2024 spotlight_

### Official Review · Reviewer_d3VV · 2023-10-30

**Soundness:** 3 good
**Presentation:** 3 good
**Contribution:** 3 good
**Rating:** 6
**Confidence:** 3

**Summary:**

This paper introduces LiDAR (Linear Discriminant Analysis Rank) as a novel  metric for assessing the quality of representations within joint embedding (JE) architectures. The authors conduct comprehensive experiments and demonstrate that their proposed LiDAR metric correlates significantly and consistently higher with downstream linear probing performance than RankMe. They further show that LiDAR demonstrates consistently strong performance in hyperparameter selection, outperforming RankMe.

**Strengths:**

- The proposed LiDAR metric for accessing the quality of representations within joint embedding (JE) architectures sounds novel.
- The theoretical motivation for proposing the LiDAR metric is clear and logical.
- Comprehensive experiments have been carried out to demonstrate the superiority of LiDAR over RankMe.
- The paper is clear and easy to follow.

**Weaknesses:**

- No disucssions on computational overhead and runtime.

**Questions:**

None.

---

> ### Author Response · Authors · 2023-11-17
> **Response to Reviewer d3VV**
>
> Thank you for carefully reading and reviewing our paper. We are glad to see that the reviewer finds the motivation clear and appreciates the comprehensive nature of our experimental results. We address the weaknesses below:
>
>  > No disucssions on computational overhead and runtime.
>
> We thank the reviewer for bringing up this point as its important for several reasons:
>
> * First is to provide the community with information on run-time overhead of LiDAR to ensure that the overhead is within bounds for the use case considered in the paper. We conduct an experiment where-in we measure the wall-clock time for running RankMe vs LiDAR for I-JEPA evaluation and report our findings in **Appendix 12** and **Table 34**. We include the table at the end of our comment for convenience. We find that the wall-clock time for evaluation on a 1GPU machine has a median value of 0.71 seconds for LiDAR which is very reasonable albeit slower than RankMe which has a median value of 0.38 seconds. The difference is is understandable given the non-optimized nature of our implementation. We find that the times reported above are much smaller than the time taken for feature extraction also reported in **Table 34**; the feature extraction times are of the order of 10s of seconds up to several minutes. Also note that standard linear probing evaluation on the ImageNet-1K dataset takes considerably longer and/or requires multiple GPUs, potential data labeling and curation [can we mention the times we observed from probing runs?].
> * The second reason why the above point raised by the reviewer is important is that our current research findings may serve as a starting point to explore the use of LiDAR directly as a training loss function (regularizer). We think this is a great direction for future work as it motivates exploring techniques from randomized numerical linear algebra (RanNLA) and beyond to ensure an efficient implementation that can work across a range of numerical precisions relevant to practice today.
>
> -----------------
> **Table 34**: Wall-clock time to calculate effective rank for \emph{RankMe} and \emph{LiDAR}. The emebddings are gathered from an I-JEPA~\citep{ijepa} checkpoint for all methods.
> | Method          | Sample count     | Median time (std. dev.) | Feature extraction |
> |-----------------|------------------|-------------------------|--------------------|
> |                 |                  | (seconds)               | (seconds)          |
> | RankMe          | 10K              | 0.38 (2.38)             | 11.29              |
> | LiDAR (Student) | 10K (n=1K, q=10) | 0.53 (0.07)             | 23.22              |
> | LiDAR (Student) | 50K (n=1K, q=50) | 0.71 (0.06)             | 96.95              |
>
> We once again thank the reviewer for raising a great question. We hope that our response addresses your concerns raised during review. We request the reviewer to provide suggestions that can help us improve the paper further so that it meets the reviewer’s bar for acceptance as a research publication.

---

> > ### Comment · Reviewer_d3VV · 2023-11-22
> >
> > Thank the authors for providing their rebuttal. I have no further questions.

---

### Official Review · Reviewer_y3Lm · 2023-10-30

**Soundness:** 4 excellent
**Presentation:** 4 excellent
**Contribution:** 3 good
**Rating:** 8
**Confidence:** 4

**Summary:**

The authors present LiDAR, a new metric designed to assess the quality of self-supervised representations in Joint Embedding (JE) architectures using the linear probing protocol, without requiring labeled downstream tasks. LiDAR employs Linear Discriminant Analysis (LDA) to evaluate dimensional collapse and implicitly integrates the SSL objective. Through thorough empirical validation, the authors demonstrate LiDAR's superiority over existing metrics in predicting optimal hyperparameters.

**Strengths:**

1) The addressed problem holds significance in self-supervised learning. By evaluating SSL representations without labeled tasks, improvements in hyperparameter selection and algorithm development are facilitated.

2) Utilizing Linear Discriminant Analysis (LDA) to assess the dimensionality collapse of SSL representations is both theoretically grounded and considerably novel. While RankMe assessed SSL methods via the effective rank of the covariance matrix (related to PCA), LiDAR focuses on the rank of the scatter-ratio matrix through LDA.

3) Detailed experiments highlight a significant and consistent positive correlation (measured by Spearman and Kendall coefficients) between LiDAR and linear probing performance on the same source-target datasets, surpassing RankMe across most settings.

**Weaknesses:**

1) The manuscript lacks evaluation on out-of-distribution (OOD) target datasets. Although the ImageNet dataset results are promising, the evidence for LiDAR's effectiveness in more realistic downstream tasks, often involving OOD datasets, is missing. Comparing with datasets like iNaturalist-2018 or Stanford Cars, following the protocol in RankMe, would strengthen LiDAR's position as a useful proxy metric.

2) LiDAR introduces two hyperparameters: the number of surrogate classes (n) and the number of samples per class (q). Yet, the manuscript does not provide guidance on determining these values. Given that these values differ across methods, a hyperparameter sensitivity analysis seems essential.

3) The comparative analysis appears limited, with LiDAR mainly being evaluated against RankMe. Other potential baselines, like $\alpha$-ReQ mentioned in related work, are overlooked.

4) In Section 4.1, the authors advocate for dimensionality reduction (DR) of embedding features, possibly adding another layer of dependency for LiDAR evaluation. The necessity of DR for LiDAR's effective application, and any specific DR algorithms used in experiments, remain ambiguous.

5) Although the authors claim LiDAR integrates the SSL objective, the manuscript does not delve into the relationship between LiDAR and different SSL losses.

**Questions:**

1) Scatter matrices, $\Sigma_w$ and $\Sigma_b$, having dimensions ( p $\times$ p ), are influenced by both n and the data dimensionality (p). In Section 4.1, the authors note they "maintain a total of 50 features" for VICreg. How does this align with rank constraints dictated by the data dimensionality?

2) What guidance can be provided for practitioners to select appropriate values for the number of surrogate classes (n) and the number of samples per class (q)?

3) The authors re-implemented RankMe (without data augmentation) for new architectures, namely I-JEPA and data2vec, using 10k images for rank estimation. This contrasts with the originally suggested 25.6k images. Meanwhile, LiDAR employs at least 50k total samples. What motivated this choice?

The rebuttal addressed the raised issues well. Therefore, I updated the review.

---

> ### Author Response · Authors · 2023-11-18
> **Response to Reviewer y3Lm (OOD dataset performance Part 1)**
>
> Thank you for carefully reading and providing a detailed review of our paper! We are very encouraged to read that the reviewer finds the paper has significance to the SSL and representation learning community and recognizes the novelty of the use of Linear Discriminant Analysis (LDA) for evaluating representations! We address the weaknesses and questions raised by the reviewers below:
>
> > The manuscript lacks evaluation on out-of-distribution (OOD) target datasets. Although the ImageNet dataset results are promising, the evidence for LiDAR's effectiveness in more realistic downstream tasks, often involving OOD datasets, is missing. Comparing with datasets like iNaturalist-2018 or Stanford Cars, following the protocol in RankMe, would strengthen LiDAR's position as a useful proxy metric.
>
> This is a very important and valid point raised by the reviewer. Fortunately, we were able to evaluate the performance of all of the SSL methods considered in our paper namely I-JEPA, data2vec, DINO, VICReg and SimCLR on 5 commonly used datasets that are considered out-of-distribution (OOD) in the SSL literature. We are excited to include them in the paper to strengthen the contribution. The datasets considered here are CIFAR10, CIFAR100, EuroSAT, Food101 and SUN397, which represent a reasonable subset of OOD datasets considered in RankMe [Garrido 2023].
>
> We follow the protocol used in RankMe [Garrido 2023] and select the most performant checkpoint for each metric (ImageNet Oracle (based on validation dataset), RankMe, augmented RankMe (where applicable) and LiDAR). These checkpoints are trained and evaluated with the datasets mentioned above. A detailed description of the evaluation protocol is in Appendix Section 10.1 of the paper. The results for the various methods are available in the following tables:
>
> - **Table 22** (average accuracies) and **23** (breakdown by dataset) for I-JEPA where the hyperparameters are set via grid search as done in I-JEPA[Assran2023]
> - **Table 24** (average accuracies) and **25** (breakdown by dataset) for I-JEPA where the hyperparameters are set via random search
> - **Table 26** (average accuracies) and **27** (breakdown by dataset) for data2vec where the hyperparameters are set via grid search
> - **Table 28** (average accuracies) and **29** (breakdown by dataset) for DINO where the hyperparameters are set via random search
> - **Table 30** (average accuracies) and **31** (breakdown by dataset) for VICReg where the hyperparameter values are copied over from RankMe[Garrido2023]
> - **Table 32** (average accuracies) and **33** (breakdown by dataset) for SimCLR where the hyperparameter values are set via random search
>
> We observe from the average accuracies that LiDAR performs better than RankMe in most cases and matches RankMe performance for a handful of cases considered in our experiments. Notably LiDAR is able to perform better than the ImageNet Oracle in OOD evaluation in certain cases ( see Table 24, Table 28, Table 30) . This observation is consistent with and  similar to the observations made in RankMe[Garrido2023] that suggests new metrics are needed to evaluate downstream performance.
>
> We have included the average linear probe accuracies calculated on the OOD datasets in the following comments for convenience. These tables and the per-dataset breakdown are also available in the updated draft
>
> **Table 22:** I_JEPA: Average linear probe accuracy recovered by \emph{RankMe} and \emph{LiDAR} on OOD datasets. Hyperparameters set via grid search.
> | Metric                            | LR                      | WD                      | Target mask scale            | Context mask scale    | Overall                 |
> |-----------------------------------|-------------------------|-------------------------|-------------------------|-------------------------|-------------------------|
> | \textcolor{gray}{ImageNet Oracle} | \textcolor{gray}{78.49} | \textcolor{gray}{78.49} | \textcolor{gray}{78.49} | \textcolor{gray}{78.49} | \textcolor{gray}{78.49} |
> | RankMe                            | 72.09                   | 75.45                   | 75.45                   | 75.15                   | 72.09                   |
> | RankMe (aug.) (Student)           | \textbf{77.78}          | \textbf{77.78}          | \textbf{77.78}          | 77.78                   |\textbf{ 77.78}                   |
> | RankMe (aug.) (Teacher)           | 72.09                   | 72.14                   | 70.20                   | 75.15                   | 72.09                   |
> | LiDAR (Student)                   | \textbf{77.78}          | \textbf{77.78}          | \textbf{77.78}          | 77.78                   | \textbf{77.78}                   |
> | LiDAR (Teacher)                   | 74.33                   | 76.84                   | 73.91                   | \textbf{78.30}          | 74.33          |

---

> ### Author Response · Authors · 2023-11-18
> **Response to Reviewer y3Lm (OOD dataset performance Part 2)**
>
> **Table 24:** I-JEPA: Average linear probe accuracy recovered by \emph{RankMe} and \emph{LiDAR} on OOD datasets. Hyperparameters set via random search.
> | \textcolor{gray}{ImageNet Oracle} | RankMe | RankMe (aug.) | RankMe (aug.) | LiDAR     | LiDAR          |
> |-----------------------------------|--------|---------------|---------------|-----------|----------------|
> |                                   |        | (Student)     | (Teacher)     | (Student) | (Teacher)      |
> | \textcolor{gray}{74.85}           | 73.24  | 73.24         | 73.24         | 76.23     | \textbf{77.34} |
>
> ----------------------------
>
> **Table 26:** data2vec: Average linear probe accuracy recovered by \emph{RankMe} and \emph{LiDAR} on OOD datasets. Hyperparameters set via grid search.
> | Metric                            | LR                      | Mask ratio              | Overall                 |
> |-----------------------------------|-------------------------|-------------------------|-------------------------|
> | \textcolor{gray}{ImageNet Oracle} | \textcolor{gray}{68.69} | \textcolor{gray}{69.07} | \textcolor{gray}{68.69} |
> | RankMe                            | 58.86                   | 58.86                   | 58.86                   |
> | RankMe (aug.) (Student)           | 58.86                   | 60.76                   | 60.76                   |
> | RankMe (aug.) (Teacher)           | 58.86                   | 60.76                   | 60.76                   |
> | LiDAR (Student)                   | \textbf{63.76}          | \textbf{73.07}          | \textbf{63.76}          |
> | LiDAR (Teacher)                   | \textbf{63.76}          | \textbf{73.07}          | \textbf{63.76}          |
>
> ----------------------------
>
> **Table 28:** DINO: Average linear probe accuracy recovered by \emph{RankMe} and \emph{LiDAR} on OOD datasets.
> | Metric                            | Teacher temperature     | Student temperature     | Overall                 |
> |-----------------------------------|-------------------------|-------------------------|-------------------------|
> | \textcolor{gray}{ImageNet Oracle} | \textcolor{gray}{84.66} | \textcolor{gray}{84.59} | \textcolor{gray}{84.66} |
> | RankMe                            | 84.66                   | 84.46                   | 84.66                   |
> | LiDAR (Student)                   | 84.66                   | 84.72                   | 84.66                   |
> | LiDAR (Teacher)                   | 84.66                   | 84.72                   | 84.66                   |
>
> ----------------------------
>
> **Table 30:** VICReg: Average Top-1 probe accuracies (across OOD datasets) computed on representation (following VISSL protocol) when tuning hyperparameters with ImageNet validation performance.
> | Method          | Cov   | inv   | LR    | WD    |
> |-----------------|-------|-------|-------|-------|
> | ImageNet Oracle | 77.83 | 77.61 | 76.94 | 76.97 |
> | RankMe          | 77.53 | 75.53 | 76.94 | 74.93 |
> | LiDAR           | 78.04 | 77.74 | 76.94 | 74.93 |
>
> ----------------------------
>
> **Table32:** SimCLR: Average Top-1 probe accuracies (across OOD datasets) computed on representation when tuning hyperparameters (random search) based on ImageNet validation performance.
>
> | Method          | Temp  | LR    | WD    |
> |-----------------|-------|-------|-------|
> | ImageNet Oracle | 80.68 | 80.84 | 80.76 |
> | RankMe          | 78.10 | 79.48 | 79.36 |
> | LiDAR           | 79.88 | 80.30 | 80.24 |
>
> ----------------------------------------------------------
>
> We were unable to use the datasets mentioned by the reviewer for the following reasons:
>
> * StanfordCars was unavailable. We were unable to download the dataset using PyTorch’s torchvision library at the time we conducted our evaluations.
> * iNaturalist-2018 due to its sheer size. Given the short turnaround during rebuttal phase, we did not have sufficient time to run experiments and share the results during the rebuttal phase.
>
> We will find a different way to download StanfordCars and run evaluations on both StanfordCars and iNaturalist-2018 for the camera-ready version as we agree with the reviewer that it is good to be as comprehensive as possible, though we feel the OOD results are already quite promising.

---

> ### Author Response · Authors · 2023-11-18
> **Response to Reviewer y3Lm (answers to weaknesses continued Part 3)**
>
> > LiDAR introduces two hyperparameters: the number of surrogate classes (n) and the number of samples per class (q). Yet, the manuscript does not provide guidance on determining these values. Given that these values differ across methods, a hyperparameter sensitivity analysis seems essential.
>
> We agree with the reviewer on the importance of having guidance on how to set the number of surrogate classes (n) and number of samples per class (q). We first note that the value of n has to be at least as large as the dimension of the features used in LDA analysis to ensure there is no artificial constraint on the rank of these feature matrices. This bound is noted in **Section 4.1 equation (6)**. For q, we note that we only need an adequate number of samples to calculate the scatter matrices. We conduct an empirical study with I-JEPA model that produces feature vectors of length 768. We choose n to be 1000 that clearly satisfies the bound mentioned above. In order to determine the sensitivity of LiDAR score on q, we conduct experiments where we vary q within the range [10, 100] and plot the results in **Figure 23**. Details are included in **Section 11** in the Appendix. We observe from **Figure 23** that the LiDAR score at q=10 is already close to 99% of its final value (and reaches beyond 99% of its final value at q=50). This quick ablation suggests that the value (n=1000, q=50) is reasonable for embeddings provided by I-JEPA.
>
> To maintain consistency across our implementations, we copy these values for data2vec that outputs embeddings of length 768 and DINO that outputs embeddings of size 256. We note that VICReg and SimCLR models output embedding vectors of length 2048. This forces us to use a larger value for n. We use a value n=5000 for SimCLR and n=10000 for VICReg while keeping q=10 for these models. We acknowledge that the reviewer raises an excellent point but due to heavy computational demands of our experiments, we were not able to perform a detailed sensitivity analysis for all methods. But we note that the heuristics used in choosing hyperparameters are already sufficient to saturate performance. We leave a careful study of these hyperparameters as future work where we may consider this question jointly with the question of LiDAR as a viable training objective.
>
> > The comparative analysis appears limited, with LiDAR mainly being evaluated against RankMe. Other potential baselines, like -ReQ mentioned in related work, are overlooked.
>
> We thank the reviewer for raising this excellent point. We agree that \alpha-ReQ is relevant to the topic of representation evaluation. However, we decided to focus on RankMe as it was shown to work better than \alpha-ReQ in-distribution and comparable OOD (and is currently the state-of-the-art approach for predicting downstream performance for joint-embedding architectures). But we will add comparisons in a future/camera-ready version of the paper.
>
> > In Section 4.1, the authors advocate for dimensionality reduction (DR) of embedding features, possibly adding another layer of dependency for LiDAR evaluation. The necessity of DR for LiDAR's effective application, and any specific DR algorithms used in experiments, remain ambiguous.
>
> We thank the reviewer for pointing out this ambiguity in our writing. First of all we apologize for the confusion caused by our statement on dimensionality reduction (DR). Our intention is not to argue for dimensionality reduction but to use the trivial bound in equation 6 in Section 4.1 as a guide to select the number of classes. We suggest that the users of LiDAR set n to be greater than the feature vector size to not impose any artificial constraint on the rank (LiDAR score). We follow this rule of thumb to select n for all of the SSL methods considered in our as explained in our answer above to the question on hyperparameter selection. We mention that dimensionality reduction could be potentially used to reduce computational complexity for cases where (p >> n) but we avoid this in our evaluations to ensure fairness. We once again apologize for any confusion.

---

> ### Author Response · Authors · 2023-11-18
> **Response to Reviewer y3Lm (answers to weaknesses Part 4 and answers to questions Part 1)**
>
> > Although the authors claim LiDAR integrates the SSL objective, the manuscript does not delve into the relationship between LiDAR and different SSL losses.
>
> We agree with the reviewer that the paper does not attempt to establish any relationship between LiDAR and various SSL objectives used in practice. We proposed LiDAR that captures the discriminative aspect of the SSL methods (using image as a class) and conjecture that the discriminative directions LiDAR is sensitive to are the same discriminative directions learned by various SSL objectives considered in the paper. We focused our efforts in this paper to thoroughly evaluate the effectiveness of LiDAR as a metric that predicts downstream performance for several SSL objectives. We leave the topic of establishing potential relationship between LiDAR and different SSL losses as a topic for future research. We thank the reviewer for their excellent suggestion as we delve deeper into this topic.
>
> **Answers to Questions raised by Reviewer y3Lm**
> > Scatter matrices, and , having dimensions ( p p ), are influenced by both n and the data dimensionality (p). In Section 4.1, the authors note they "maintain a total of 50 features" for VICreg. How does this align with rank constraints dictated by the data dimensionality?
>
> We thank the reviewer for the question. We apologize to the reviewer for the confusion caused above due to multiple typos in our initial draft. We note that the above (n=5K, q=10) LiDAR hyperparamters are used for SimCLR leading to a dataset with 50000 (and not 50) features. Secondly, we use (n=10K, q=10) for VICReg as noted in Section 9.1 in the Appendix of the original draft. We have updated our draft to remove any confusion. Once gain we apologize to the reviewer for causing confusion in our presentation.
>
> The value of n are chosen for various SSL objectives to ensure that the number of surrogate classes exceeds the length of feature vectors (p). Thus, our choice of n ensures that we do not impose any artificial constraints on the rank estimated by LiDAR.
>
> > What guidance can be provided for practitioners to select appropriate values for the number of surrogate classes (n) and the number of samples per class (q)?
>
> (We repeat our answer from a related question above)
> We thank the reviewer for raising this important question of how to set number of surrogate classes (n) and number of samples per class (q). We first note that the value of n has to be at least as large as the dimension of the features used in LDA analysis to ensure there is no artificial constraint imposed by LiDAR on these feature vectors. This bound is noted in Section 4.1 equation (6). For q, we note that we only need an adequate number of samples to calculate the scatter matrices. We conduct an empirical study with I-JEPA model that produces feature vectors of length 768. We choose n to be 1000 that clearly satisfies the bound mentioned above. In order to determine the sensitivity of LiDAR score on q, we conduct experiments where we vary q from [10, 100] and plot the results in Figure 23 and include the details in Section 11 in the Appendix. We can observe from Figure 23 that the LiDAR score is already close to 99% of its final value at q=10 and reaches more than 99% of its value for q=50. This quick ablation suggests that the value (1000, 50) is reasonable for embeddings provided by I-JEPA.
>
> To maintain consistency across our implementations we copy these values for data2vec that outputs embeddings of length 768 and DINO that outputs embeddings of size 256. We note that VICReg and SimCLR models output embedding vectors of length 2048 which forces us to use a larger value for n. We use a value n=5000 for SimCLR and n=10000 for VICReg while keeping q=10 for these models. While a sensitivity study for each method would be ideal, we note that the heuristics used to choose hyperparameters are already sufficient to saturate performance. We leave a careful study of these hyperparameters as future work where we may consider this question jointly with making LiDAR a viable training objective.

---

> ### Author Response · Authors · 2023-11-18
> **Response to Reviewer y3Lm (answers to questions Part 2)**
>
> > The authors re-implemented RankMe (without data augmentation) for new architectures, namely I-JEPA and data2vec, using 10k images for rank estimation. This contrasts with the originally suggested 25.6k images. Meanwhile, LiDAR employs at least 50k total samples. What motivated this choice?
>
> The reviewer is correct on using 10000 samples for our rank estimation. Our choice was motivated by a rank convergence analysis for RankMe we conducted (using I-JEPA as our sample architecture which is not covered in RankMe [Garrido 2023]). **Figure 24** summarizes the outcome of the analysis. We observe that we get to greater than 99% of the rank estimate at 25600 samples with just using 10000 samples. This gives us confidence that the number of samples used in our experiments is reasonable. To complete the story we will take the reviewer’s feedback and rerun our analysis with 25600 samples for the camera-ready version of the paper.
>
> **To ensure fairness in our empirical analysis, we evaluate a version of RankMe that we call augmented-RankMe (basically, RankMe evaluated on the same dataset created to evaluate LiDAR).** The results of these comparisons are reported **Table 1, Table 2, Table 3 and Table 4** for **I-JEPA** and in **Table 5 and Table 6** for **data2vec**. We observe from these tables that LiDAR shows better performance both in terms of correlation and linear probe accuracy recovered for various hyperparameter settings used in our experiments. We thank the reviewer for their questions and will address any remaining feedback in the camera-ready version of the paper.

---

> > ### Comment · Reviewer_y3Lm · 2023-11-19
> >
> > I thank the authors for the detailed rebuttal. If there are follow-up questions, I will post a follow-up comment.

---

> > > ### Comment · Reviewer_y3Lm · 2023-11-22
> > >
> > > A follow-up question on some detail:
> > >
> > > The choice of 10k samples for rank estimation for RankMe (baseline) motivated by the convergence analysis (new experiments in Appendix 13): The figure shows that RankMe estimation improves as the number of samples increases to the amount recommended in the original paper (25.6k). What is the motivation for choosing a lower amount (at 99% of rank quality), which probably leads to lower results? What is the performance obtained at 100% quality (25.6 samples)? In the ablation of LiDAR (new experiments in the Appendix 11), we observe the choice of the minimum amount of samples (1000 x 50) that gives the maximum rank quality.

---

> ### Author Response · Authors · 2023-11-22
> **Impact of number of samples on RankMe**
>
> We thank the reviewer for their continued engagement with our paper and for their very insightful questions.
>
> > The choice of 10k samples for rank estimation for RankMe (baseline) motivated by the convergence analysis (new experiments in Appendix 13): The figure shows that RankMe estimation improves as the number of samples increases to the amount recommended in the original paper (25.6k). What is the motivation for choosing a lower amount (at 99% of rank quality), which probably leads to lower results?
>
> We first note here that we use 25600 samples in our analysis of SimCLR, ViCReg and DINO in our paper.
>
> The choice of 10000 samples applies to I-JEPA and data2vec which are new methods considered in our paper. Our choice of 10000 is motivated in part by the fact that I-JEPA and data2vec output smaller feature vectors compared to the methods in RankMe paper, 768 vs 2048 dimensions and by our ablation in Appendix 13 suggests that 10000 samples can capture 99% of maximum rank. Our conjecture here is that the shape of probe accuracy versus RankMe plot (scatterplot) is important and not necessarily the raw numbers for RankMe and went with 10000 samples in our analysis.
>
> > What is the motivation for choosing a lower amount (at 99% of rank quality), which probably leads to lower results? What is the performance obtained at 100% quality (25.6 samples)?
>
> The reviewer raises an excellent concern that we share on our side as well. To understand the impact of the number of samples on our analysis, we ran an ablation where we use 25600 for vanilla RankMe and report the results in Appendix 14. Our results suggest that the differences in Kendall's-Tau correlation is minimal. More importantly, there are no changes in the linear probe accuracies recovered with in-domain data (Imagenet-1K) and the OOD performance remains the same as the checkpoints identified by the analysis are the same. We thank the reviewer for pushing for this analysis as it's scientifically useful to understand the properties of RankMe (and other methods) used in this work.
>
> -----------------------------------------
>
> We include the tables for the number of samples ablation with I-JEPA and data2vec below:
>
> **Table 35:** I-JEPA: Kendall's $\tau$ coefficient between effective ranks of \emph{RankMe} and linear probe accuracy. \emph{RankMe} estimated with  10000 samples and 25600 samples. Hyperparameters are varied via grid search.
> | Hyperparameter     | 10000 samples | 25600 samples |
> |--------------------|---------------|---------------|
> |                    | (RankMe)      | (RankMe)      |
> | Learning rate      | 0.6835        | 0.6835        |
> | Weight decay       | 0.5040        | 0.5050        |
> | Target mask scale  | 0.2867        | 0.2847        |
> | Context mask scale | 0.4246        | 0.4256        |
> | Overall            | 0.5830        | 0.5829        |
>
> **Table 36:** I-JEPA: Kendall's $\tau$ coefficient between effective ranks of \emph{RankMe} and linear probe accuracy. \emph{RankMe} estimated with  10000 samples and 25600 samples. Hyperparameters are varied via random sampling.
> | Hyperparameter | 10000 samples | 25600 samples |
> |----------------|---------------|---------------|
> |                | (RankMe)      | (RankMe)      |
> | Overall        | 0.8314        | 0.8308        |
>
> **Table 37:** data2vec: Kendall's $\tau$ coefficient between effective ranks of \emph{RankMe} and linear probe accuracy. \emph{RankMe} estimated with  10000 samples and 25600 samples. Hyperparameters are varied via grid sampling.
> | Hyperparameter | 10000 samples | 25600 samples |
> |----------------|---------------|---------------|
> |                | (RankMe)      | (RankMe)      |
> | Learning rate  | 0.2410        | 0.2392        |
> | Mask ratio     | 0.2683        | 0.2706        |
> | Overall        | 0.2238        | 0.2216        |
>
> Furthermore, **Figure 25**, **Figure 26** and **igure 27** in **Appendix 14** shows a scatter plot of probe accuracy vs RankMe calculated with 10000 and 25600 samples. These figures show that the differences in RankMe values are small and the impact on our analysis with in-domain and OOD is negligible.
>
> ---------------------------------------------
> We once again thank the reviewer for pushing for this analysis as it's scientifically useful to understand the properties of RankMe (and other methods) used in this work. We hope that our response addresses your concerns raised during review. We request the reviewer to provide suggestions that can help us improve the paper further so that it meets the reviewer’s bar for acceptance as a research publication.

---

### Official Review · Reviewer_bVwZ · 2023-10-30

**Soundness:** 3 good
**Presentation:** 2 fair
**Contribution:** 3 good
**Rating:** 6
**Confidence:** 1

**Summary:**

This paper proposes a new metric for the measurement of the representation quality within joint embedding architectures, which is called Linear Discriminat Analysis Rank.
This metric discriminate between informative and uninformative features by quantifying the rank of the Linear Discriminant Analysis matrix.
The experiments on several downstream tasks show the proposed metric improves the performance.

**Strengths:**

1. The paper generally explain their motivation and inspiration clearly.
2. The experiments are sufficient to show the effectiveness of the proposed method on the downstream tasks.
3. The supplementary material shows the details of the proposed method.

**Weaknesses:**

1. The abbreviation of the proposed method "LiDAR" is irrelavent of the problem it tries to solve. The irrelavent abbreviation misleads the readers what the paper want to express. Some readers may think the paper is on the lidar.

2. Some typos:

a. In section 2.1: In practice, a set of downstream tasks {T_j} are used to asses ->  In practice, a set of downstream tasks {T_j} are used to assess

b. in Section 4.0.1: There is a blank line, where a sentance seem missed.

**Questions:**

Actually, I have no background of the topics of the paper. I cannot assess and ask any questions on this paper. I do not know why this paper is assigned to me. I think the reason maybe the abbreviation of the proposed method "LiDAR" is related with my previous paper, however, this paper is not related with lidar at all.

---

> ### Author Response · Authors · 2023-11-17
> **Response to Reviewer bVwZ**
>
> Thank you for your time and effort in reviewing our paper.
>
> > The abbreviation of the proposed method "LiDAR" is irrelavent of the problem it tries to solve. The irrelavent abbreviation misleads the readers what the paper want to express. Some readers may think the paper is on the lidar.
>
> We recognize the potential for confusion which can often occur when producing names for methods that are intended to be memorable and creative, which we feel strongly about. Therefore, to reduce potential for confusion, we expand the term LiDAR in the abstract to **Li**near **D**iscriminant **A**nalysis **R**ank. We hope that along with the title referring to the goal of the paper this makes it clear to future readers that LiDAR refers to a metric for evaluating representations of joint embedding methods where the name is meant to be a playful way to refer to ‘sensing’ representation quality, as mentioned in the title.
>
> > a. In section 2.1: In practice, a set of downstream tasks {T_j} are used to asses → In practice, a set of downstream tasks {T_j} are used to assess
>
> > b. in Section 4.0.1: There is a blank line, where a sentance seem missed.
>
> We thank the reviewer for carefully reading our paper and bringing the above issues to our notice. We have fixed the issues in the current revision of our draft. We commit to fixing any issues that remain in a camera-ready version of our paper and will ensure the paper goes through a round of proofreading and spellcheck.

---

### Author Response · Authors · 2023-11-18
**General Response to All Reviewers**

We thank all reviewers for their time and thoughtful comments. We have taken the feedback provided by the reviewers very seriously and used it to improve our paper. Major changes include newly added appendices described below:

- Appendix 10 that includes results from OOD evaluations
-  Appendix 11 that has an ablation that studies the impact of (n, q) hyperparameters for LiDAR using I-JEPA as an example
-  Appendix 12 includes information on the run-time complexity of LiDAR and RankMe using I-JEPA as an example
-  Appendix 13 that studies the impact of the number of samples for estimating RankMe with I-JEPA which is an example of a new architecture considered in this work.
-  Fixed typos and rewrote portions of Section 4.1 on Compute Considerations

Additionally, we have fixed numerous errors and typos in our draft pointed out by the reviewers as well as those that we discovered in our review of the paper. These changes have been uploaded to OpenReview as a rebuttal revision.
We close our general repose by noting that we commit to fixing any issues that remain in a camera-ready version of our paper and will ensure the paper goes through a round of proofreading and spellcheck.

We respond to specific comments and answer questions raised by individual reviewers in the following.

---

### Author Response · Authors · 2023-11-22
**Updated version**

We updated our paper using the feedback from the reviews below. Specifically we added Appendix 14 based on questions raised by Reviewer **y3Lm**. We also corrected several typos and copy-paste issues in the updated manuscript as well. We thank the reviewers spending their time carefully reading our manuscript. We hope that our response addresses the reviewers concerns raised during review. We invite each reviewer to provide suggestions that can help us improve the paper further so that it meets the reviewer’s bar for acceptance as a research publication.

---

### Meta-Review · Area_Chair_djQ7 · 2023-12-11

**Metareview:**

The paper addresses the problem of measuring representation quality in self-supervised learning, particularly in the context of joint embedding.

Strengths:
 - A new method, which advances theoretically and practically on the current RankMe approach.
 - Post-rebuttal, good evaluations on some OOD datasets

Weaknesses:
 - The authors note that it remains a goal to evaluate on StanfordCars and iNaturalist.  It can be argued that such evaluations ought to be in the original paper, so it remains a valid weakness, independently of whether the rebuttal time period was sufficient to perform the evaluations.

**Justification For Why Not Higher Score:**

As mentioned, the paper is making an improvement over existing work, namely RankMe.  It is therefore expected to perform evaluations on at least the same datasets in the existing work.

**Justification For Why Not Lower Score:**

The reviewers consider the work a useful and novel development in the JE-SSL domain, which is a core subfield to range of ICLR attendees.

---

### Decision · Program_Chairs · 2024-01-16

Accept (spotlight)